# RELEASE THE POWERS OF PROMPT TUNING: CROSS-MODALITY PROMPT TRANSFER

**Ningyuan Zhang, Jie Lu,**[*] **Keqiuyin Li, Zhen Fang, Guangquan Zhang**
Australian Artificial Intelligence Institute, University of Technology Sydney
ningyuan.zhang@student.uts.edu.au
{jie.lu, keqiuyin.li, zhen.fang, guangquan.zhang}@uts.edu.au

## ABSTRACT

Prompt Tuning adapts frozen models to new tasks by prepending a few learnable embeddings to the input. However, it struggles with tasks that suffer from data scarcity. To address this, we explore *Cross-Modality Prompt Transfer*, leveraging prompts pretrained on a data-rich modality to improve performance on data-scarce tasks in another modality. As a pioneering study, we first verify the feasibility of cross-modality prompt transfer by directly applying frozen source prompts (trained on the source modality) to the target modality task. To empirically study cross-modality prompt transferability, we train a linear layer to adapt source prompts to the target modality, thereby boosting performance and providing ground-truth transfer results. Regarding estimating prompt transferability, existing methods show ineffectiveness in cross-modality scenarios where the gap between source and target tasks is larger. We address this by decomposing the gap into the modality gap and the task gap, which we measure separately to autonomously select the best source prompt for a target task. Additionally, we propose *Attention Transfer* to further reduce the gaps by injecting target knowledge into the prompt and reorganizing a top-transferable source prompt using an attention block. We conduct extensive experiments involving prompt transfer from 13 source language tasks to 19 target vision tasks under three settings. Our findings demonstrate that: (i) cross-modality prompt transfer is feasible, supported by in-depth analysis; (ii) measuring both the modality and task gaps is crucial for accurate prompt transferability estimation, a factor overlooked by previous studies; (iii) cross-modality prompt transfer can significantly release the powers of prompt tuning on data-scarce tasks, as evidenced by comparisons with a newly released prompt-based benchmark.

## 1 INTRODUCTION

As pretrained Transformers (Vaswani et al., 2017) become larger, the demand for transferring them to new tasks in a parameter-efficient way grows heavier (Peng et al., 2024). Prompt Tuning (Lester et al., 2021) emerges as a leading Parameter-Efficient Transfer Learning (PETL) method, updating only the learnable vectors prepended to model inputs. It makes the least modifications to the model's architecture compared to other PETL approaches (Guo et al., 2022), making it particularly compatible with complex or blackbox Transformer models, thereby warranting increasing popularity. However, some modalities struggle to benefit from prompt tuning. Data-scarce modalities, for example, lack sufficient training samples to fully optimize the prompt (Su et al., 2022).

Our research seeks to tackle the challenges by exploring *Cross-Modality Prompt Transfer*, aiming to release the powers of prompt tuning on data-scarce tasks by transferring prompts pretrained on Natural Language Processing (NLP) tasks, which are well-suited for prompt tuning due to their data-rich nature (Lu et al., 2022). Such exploration can greatly benefit data-scarce modalities by leveraging the rich data resources and well-established frameworks of the text modality through the transfer of text-pretrained prompts, thereby unlocking the power of pretrained models and mitigating the challenges these modalities face.

---

[*]Corresponding Author

In the literature, prompt transfer has been proven effective between NLP tasks (Vu et al., 2022; Su et al., 2022). We hypothesize that such practice can be safely extended to cross-modality scenarios and conduct extensive studies to verify our hypothesis. We start with proving the feasibility of cross-modality prompt transfer and empirically studying the prompt transferability in cross-modality scenarios by designing two transfer settings: *Frozen Prompt Transfer* and *Projection Transfer*. Frozen prompt transfer directly applies text-pretrained prompts (source prompts) to non-text tasks without modification, testing whether any transferability exists. For projection transfer, it enhances the performance of the frozen source prompts and obtains the ground-truth transfer results by training a linear layer to adapt to the target tasks that linguistic knowledge stored in the frozen source prompts.

With the ground-truth transfer results, we further study what determines the prompt transferability in cross-modality scenarios by analyzing the gap between source and target tasks. Observing that existing methods for in-modality scenarios encounter limitations, we hypothesize that the gap between the source and target task is more complex in cross-modality scenarios. In response, we identify two gaps a source prompt must overcome to be employed on a different modality task: (i) the modality gap formed by the differences in data distributions and models between the source and target modalities, and (ii) the task gap formed by the differing nature of the tasks themselves. By measuring these gaps, we propose a novel prompt transferability estimation approach that autonomously select the best source prompt more accurately compared to current methods designed for in-modality scenarios.

Based on the analysis of the modality gap and task gap, we further explore whether the performance of prompt tuning can be further boosted by bridging these gaps. In response, *Attention Transfer* is proposed. It reduces the gaps by injecting target knowledge into a highly transferable source prompt and by an attention block that adapts the source prompt more effectively. As a result, attention transfer can boost prompt tuning to a level comparable to or even better than the newest prompt tuning benchmark, demonstrating the potential of cross-modality prompt transfer. In summary, our contributions can be summarized as:

- We explore cross-modality prompt transfer as an effective approach for boosting prompt tuning and verify the feasibility through extensive experiments and in-depth analysis, addressing a critical research gap and opening new possibilities for data-scarce modalities.

- We introduce a novel method for estimating prompt transferability by quantifying the modality and task gaps, which the existing in-modality methods have overlooked, as the gaps are enlarged in the cross-modality scenario. As a result, our method offers a more accurate solution compared to existing in-modality methods.

- We further demonstrate the powers of cross-modality prompt transfer through *Attention Transfer*, which eases the modality gap and task gap by injecting target knowledge into the prompt and utilizing a top-transferable source prompt more effectively. As a result, attention transfer enables prompt tuning to perform comparably or even better than the best prompt-based benchmark.

## 2 RELATED WORK

**Parameter-Efficient Transfer Learning.** As model size increases, finetuning becomes infeasible for adapting pretrained models to new tasks. Therefore, PETL approaches that steer a pretrained model by tuning only a small amount of weights are favored, as they help avoid storing different model instances for individually varied tasks (Ding et al., 2023). Most PETL methods are built upon three baselines: (i) Adapter (Houlsby et al., 2019) that inserts learnable projectors between layers, (ii) Low-Rank Adaptation (LoRA) (Hu et al., 2021) that trains rank decomposition matrices as the update matrices for model weights, and (iii) Prompt Tuning (Lester et al., 2021) that concatenates learnable embeddings with input embeddings. Among these baselines, prompt tuning acquired many of popularity and had been extended to various pretrained models (Wang et al., 2024a) due to its ability to steer architecturally complex or black-box models, as it makes the least modifications to the model architecture. As an influential follow-up study, Jia et al. (2022) extended prompt tuning to the Computer Vision (CV) domain and proposed Visual Prompt Tuning (VPT), integrating prompt tuning with Vision Transformers (Dosovitskiy et al., 2020). Following VPT, Wang et al. (2024b) explored the influence of prompt initialization to VPT: after empirical verification that the mutual information between prompt and image patch embeddings tends to increase as prompt tuning proceeds, they proposed Self-Prompt Tuning (SPT) that initializes the prompts with sampled image

Figure 1: **(a) Source Prompt Tuning**: Perform prompt tuning on NLP tasks with RoBERTa to get the source prompts. **(b) Frozen Prompt Transfer**: Source prompts are used as-is on target vision tasks to verify the feasibility of cross-modality prompt transfer. **(c) Projection Transfer**: Source prompts are projected and prepended to the patch embeddings, to adapt the source knowledge.

patch embeddings. SPT demonstrated that VPT can be boosted significantly if properly initialized and achieved a new benchmark for prompt-based PETL methods. Our investigated cross-modality prompt transfer can be regarded as initializing the prompt with adapted NLP-pretrained prompt. It and SPT both focus on prompt initialization but are from different perspectives. Therefore, to demonstrate the effectiveness of cross-modality prompt transfer, SPT is chosen for comparison.

**Prompt Transfer.** Although parameter-efficient, prompt tuning still falls below full-finetuning. Moreover, training a prompt is often slower than finetuning and prompts tend to be less stable during optimization (Li & Liang, 2021). Therefore, researchers resort to prompt transfer to boost the performance and promote stable optimization for prompt tuning. Prompt transfer first trains a source prompt on a source task, before it initializes the target prompt with the trained source prompt for a target task. Vu et al. (2022) first explored prompt transfer between different NLP tasks and used the similarity between prompts to measure the prompt transferability the different tasks share. In addition, Su et al. (2022) explored prompt transfer between different language models and used a projection module to mitigate the discrepancy resulting from different language models. They also used the overlapping rate of activated neurons in the language model to measure the prompt transferability, which is more effective than the similarity measurement. Vu et al. (2022) and Su et al. (2022) both verified that prompt tuning can benefit from pretraining the prompts on an intermediate task. However, whether the conclusion still holds if the intermediate task comes from a different modality remains underexplored. In response, we explore the prompt transfer between not only different models but also different modalities, filling the gaps left by previous studies.

## 3 METHODS

### 3.1 CROSS-MODALITY PROMPT TRANSFER

**Visual Prompt Tuning.** Before diving into cross-modality prompt transfer, the basic concept of VPT needs to be introduced. VPT extends the concept of prompt tuning (Lester et al., 2021) from language to vision and serves as a representative PETL method for ViT (Dosovitskiy et al., 2020). VPT requires a target vision task $\mathcal{T}_t = \{x_t, y_t\}$ and a pretrained ViT consisting of an image embedder $E_v$ and a backbone $B_v$. The goal of VPT is to minimize the empirical error on the target vision task:

$$\arg\min_{p, H_v} \mathcal{L}\left[H_v \circ B_v(p \parallel E_v(x_t)), y_t\right] \tag{1}$$

Where $p$ represents the prompt embeddings, $H_v$ represents the classification head, $\circ$ and $\parallel$ stand for function and vector concatenation, respectively. Learnable modules are colored in orange.

**Frozen Prompt Transfer.** To verify the feasibility of cross-modality prompt transfer, frozen source prompts $p_s$ are prepended to the patch embeddings and fed to the ViT (Figure 1b). Linear probing (Oord et al., 2018) is performed on top of the [CLS] features to see if a frozen source prompt can help ViT form improved feature clusters, compared to a randomly initialized prompt and vanilla linear probing where no prompts are included. In frozen prompt transfer, the only trainable module is the

output predictor $H_v$, leaving the learning objective in Equation 1 to be reformulated into:

$$\arg\min_{H_v} \mathcal{L}\left[H_v \circ B_v(p_s \parallel E_v(x_t)), y_t)\right] \tag{2}$$

**Projection Transfer.** To adapt the linguistic knowledge stored in the source prompt to CV tasks, source prompts are kept frozen but passed to a learnable one-layer linear projector $P$ before being prepended to the image embeddings (Figure 1c). During the transfer, only the output predictor $H_v$ and the projector $P$ are trainable, reformulating objective 1 into:

$$\arg\min_{H_v, P} \mathcal{L}\left[H_v \circ B_v(P(p_s) \parallel E_v(x_t)), y_t\right] \tag{3}$$

In the literature, Merullo et al. (2022) verify that the semantic spaces of language and vision can be channeled simply through a linear projector. It, therefore, becomes interesting to explore in cross-modality prompt transfer, whether the source and target prompt spaces can be connected via a linear layer, the use of which can also be regarded as helping source prompts overcome both the modality and task gaps: smaller gaps would generally yield better performance. Therefore, projection transfer is investigated to study cross-modality prompt transferability.

### 3.2 PROMPT TRANSFERABILITY ESTIMATION

Prompt transferability estimation aims to select the best source prompt for a given target task without going through repetitive prompt tuning processes with all the source prompts. Existing methods achieve this by calculating the cosine similarity (Vu et al., 2022) or model activation similarity (Su et al., 2022) between the source and target prompts. However, these methods are designed for the in-modality scenario, where the modality gap is negligible and the task gap introduces the main discrepancy between source and target tasks. Their effectiveness encounters limitations in the cross-modality scenario as the modality gap is enlarged. Therefore, in estimating the cross-modality prompt transferability, the key lies in *measuring the modality gap and task gap*.

**Modality Gap.** The modality gap is interpreted as the distribution difference between source and target data that comes from different data types. The difference between source and target models and modalities causes it. In practice, directly estimating the distribution gap between source and target data is non-realistic because: (i) it is computationally inefficient as some tasks might contain tons of samples, and (ii) different source datasets have similar data embeddings because they are sampled from the token embeddings of the same language model, leading to insufficiently distinct distributional differences between all the source datasets and a particular target dataset. The source prompts, on the other hand, can effectively exploit and condense information from the source data as stated by Lester et al. (2021), and Zhong et al. (2021). Therefore, the source prompt is treated as an abstraction of the source data which it will replace to participate in calculating the modality gap. As a result, the modality gap $\mathcal{G}_M$ between a source NLP task $s$ and a target CV task $t$ is estimated as the Maximum Mean Discrepancy (MMD) (Gretton et al., 2012) between the source prompt and target image patch embeddings (more details can be found in the Appendix):

$$\mathcal{G}_M(s, t) = MMD(p_s, E_v(x_t)) \tag{4}$$

**Task Gap.** Existing prompt transferability estimation methods (Vu et al., 2022; Su et al., 2022) are designed for in-modality scenarios where the modality gap is negligible. Therefore, they can be regarded as estimating the task gap by measuring the similarity among prompts trained on different tasks, treating the trained prompts as task embeddings. However, in cross-modality scenarios, directly measuring the similarity between source and target prompts could lead to poor estimation of the task gap, as the source and target prompts lie in different semantic spaces. To this end, we propose a simple yet effective method that trains a universal linear projector to cast the source prompts to the target prompt space, enabling existing methods to function under the cross-modality scenario.

Specifically, given a target CV task, we first perform vanilla VPT to obtain the target prompt. Then, source prompts trained on different NLP tasks are shuffled with each other, after which they are sent to the universal projector $P_u$ which consists of a single linear layer. $P_u$ is learned by minimizing the Euclidean distance between the shuffle-then-projected source prompts and the target prompt (more details can be found in the Appendix). Note that, although they share the same architecture,

the universal projector $P_u$ used here is functionally different from the linear projector $P$ used in projection transfer. $P$ is learned from scratch in every source-target pair (works only on this pair) and its goal is to minimize the target error, but $P_u$ is learned from every target task (works on every source prompt) to minimize the vector distance.

After $P_u$ is trained, the task gap $\mathcal{G}_T$ between an NLP task $s$ and the given CV task $t$ is measured by:

$$\mathcal{G}_T(s,t) = -cos(\frac{1}{l}\sum_i^l [P_u(p_s^i)], \frac{1}{l}\sum_i^l p_t^i) \tag{5}$$

Where $cos(\cdot,\cdot)$ represents the cosine similarity between two vectors, $p_s^i$ and $p_t^i$ are the i-th vector of the source and target prompt respectively, and $l$ is the number of vectors in the source and target prompt.

**Prompt Transferability.**   With the modality gap $\mathcal{G}_M$ and task gap $\mathcal{G}_T$ defined, the cross-modality prompt transferability between a source NLP task $s$ and a target CV task $t$ is then estimated by combining $\mathcal{G}_M$ and $\mathcal{G}_T$:

$$T(s,t) = -(\mathcal{G}_M(s,t) + \mathcal{G}_T(s,t)) \tag{6}$$

### 3.3   ATTENTION TRANSFER

To further help the source prompts overcome both the modality and task gaps, we propose a novel cross-modality prompt transfer scheme named attention transfer (Figure 2). It eases the modality gap by selecting a source prompt with a small MMD value to the target data, and by injecting target knowledge into the final prompt. For the task gap, it uses a *Prompt Concentrator* ($C$) to elicit the source knowledge from the source prompts in a more complex but effective way.

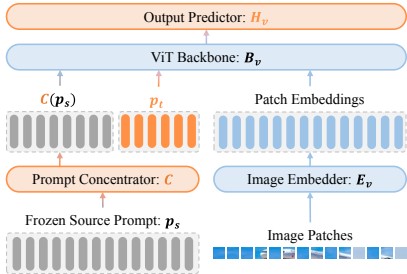

Figure 2: **Attention Transfer**.

The prompt concentrator takes the frozen source prompt $p_s$ (length $l$) as input and outputs a concentrated and adapted source prompt of length $l_s$ ($l_s < l$) by the attention mechanism. It has three key components: (i) the query $Q \in \mathbb{R}^{l_s \times d_{hs}^v}$ that decides the importance of each vector in the source prompt, (ii) the key projection matrix $W_k$ that projects the source prompt to the space of $Q$ for query operation, and (iii) the value projection matrix $W_v$ that adapts the source prompt to the target task. $W_k$ and $W_v$ are in $\mathbb{R}^{d_{hs}^l \times d_{hs}^v}$, where $\mathbb{R}^{d_{hs}^{l/v}}$ is the hidden dimension of the language/vision model. The output is obtained by the conventional attention operation:

$$C(p_s) = softmax(Q \times (p_s \times W_k)^T) \times (p_s \times W_v) \tag{7}$$

The prompt concentrator not only better arranges the source prompt, but also makes space for target knowledge. After the source prompt is concentrated, it is concatenated with a sequence of learnable prompts (randomly initialized, termed the target prompt) to form the final prompt for the target task. These target prompts aim at learning pure target knowledge, with a length of $l_t$ ($l_t = l - l_s$).

In attention transfer, the trainable modules are the prompt concentrator, target prompt, and output predictor, with the following learning objectives:

$$\underset{H_v,C,p_t}{\arg\min} \mathcal{L}\left[H_v \circ B_v(C(p_s) \parallel p_t \parallel E_v(x_t)), y_t\right] \tag{8}$$

In the literature, Wang et al. (2023) verified that prompt tuning the query, key, and value in an attention block can help a pretrained model learn new attention patterns required by a target task. Our attention block has similar aims but differs in detail: it also adapts pretrained modules to target tasks through learning new attention patterns. But it only involves the query prompt, as its key and value matrices are not frozen, enabling new attention patterns to be grasped without key and value prompts.

## 4 EXPERIMENTS

### 4.1 EXPERIMENTAL SETUP

To study cross-modality prompt transfer thoroughly, a wide range of NLP tasks and CV tasks are involved. Source prompts are trained using a pretrained language model RoBERTa (Liu et al., 2019) on each NLP task with a uniform length of 100. On target CV tasks, a pretrained ViT is used. The final prompt length is kept at 100 for all transfer scenarios and vanilla VPT.

ViT is a classic and representative Transformer-based vision model, and VPT was initially experimented on ViT. Therefore, ViT is selected as the target vision model. RoBERTa is an encoder language model that shares nearly the same architecture to ViT. Thus, it might share a similar prompt space to ViT. Therefore, RoBERTa is chosen as the source language model. Note: (i) Only the base-model sizes are explored (e.g. RoBERTa-base and ViT-base which all have a weight amount of $\sim$125M). (ii) For ViT, the `vit-base-patch16-224-in21k` pretrained checkpoint is explored. This ViT checkpoint is pretrained in a supervised manner on ImageNet-21k (Deng et al., 2009), with an image size of $224 \times 224$ and a patch size of $16 \times 16$. The exploration of different model sizes, different source language models, and diverse ViT pretraining schemes will be left to future work.

**Source NLP Tasks.** In total, 13 investigated NLP tasks are selected as the source tasks to train the source prompt. These tasks can be categorized into four groups: (i) Sentiment Analysis (SA) tasks, including IMDB (Maas et al., 2011), SST-2 (Socher et al., 2013), laptop (Pontiki et al., 2016), restaurant (Pontiki et al., 2016), Movie Rationales (Movie, Zaidan et al. (2008)), and TweetEval (Tweet, Barbieri et al. (2020)). (ii) Natural Language Inference (NLI) tasks, including MNLI (Williams et al., 2017), QNLI (Wang et al., 2018), and SNLI (Bowman et al., 2015). (iii) Ethical Judgment (EJ) tasks, including deontology and justice (Hendrycks et al., 2020). (iv) Paraphrase Identification (PI) tasks, including QQP (Sharma et al., 2019) and MRPC (Dolan & Brockett, 2005).

**Target Vision Tasks.** The VTAB-1K (Zhai et al., 2019) image classification tasks are chosen as the target tasks. VTAB-1K consists of 19 diverse image classification tasks, each with a training set of 1000 images. These tasks can be divided into three main categories: (i) Natural tasks (including CIFAR100 (Krizhevsky et al., 2009), Caltech101 (Li et al., 2004), DTD (Cimpoi et al., 2014), Flowers102 (Nilsback & Zisserman, 2006), Pets (Parkhi et al., 2012), SVHN (Netzer et al., 2011), and SUN397 (Xiao et al., 2010)) that contain natural images captured using standard cameras; (ii) Specialized- tasks (including Patch Camelyon (Veeling et al., 2018), EuroSAT (Helber et al., 2018), Resisc45 (Cheng et al., 2017), and Retinopathy (Dugas et al., 2015)) that contain images captured via specialized equipment; and (iii) Structured tasks (including Clevr (Johnson et al., 2017), DMLab (Zhai et al., 2019), KITTI (Geiger et al., 2012), dSprites (Matthey et al., 2017), and SmallNORB (LeCun et al., 2004)) that require geometric comprehension like object counting.

**Implementation Details.** For vanilla linear probing and frozen prompt transfer, the original VPT repository released a set of linear probing hyperparameters that were carefully grid-searched on each CV task. The same hyperparameters are adopted for vanilla linear probing and frozen prompt transfer, as the nature of the frozen prompt transfer is identical to that of linear probing.

For vanilla VPT, we follow the procedure of Jia et al. (2022) performing grid-search on learning rates $\{0.01, 0.05, 0.1, 0.25, 0.5, 1, 2.5, 5, 10\}$ and weight decay values $\{0, 0.0001, 0.001, 0.01\}$, with a batch size of 64, warm-up steps of 10, cosine learning rate scheduler, and an SGD optimizer with a momentum of 0.9. Note that following Jia et al. (2022), the learning rate is multiplied by (Batch Size/256) before training starts. We use the official 800-200 split released by Zhai et al. (2019) to perform the grid-search, training on 800 images and validating using the remaining 200.

For projection transfer, a batch size of 64, a learning rate of 0.005, and a weight decay of 0.001 are used. For the optimizer, Adam (Kingma & Ba, 2014) is adopted.

For attention transfer, we first perform a grid-search to find the potentially best source prompt and its concentrated length $l_s$. Note that the source prompt is searched only from the top-transferable ones and the hyperparameters of projection transfer are adopted during this stage. Then, we perform another round of grid-search using the potentially best source prompt and $l_s$ to find the potentially best learning rate from $\{1, 0.5, 0.1, 0.05, 0.01, 0.005, 0.001, 0.0005\}$ and weight decay value from

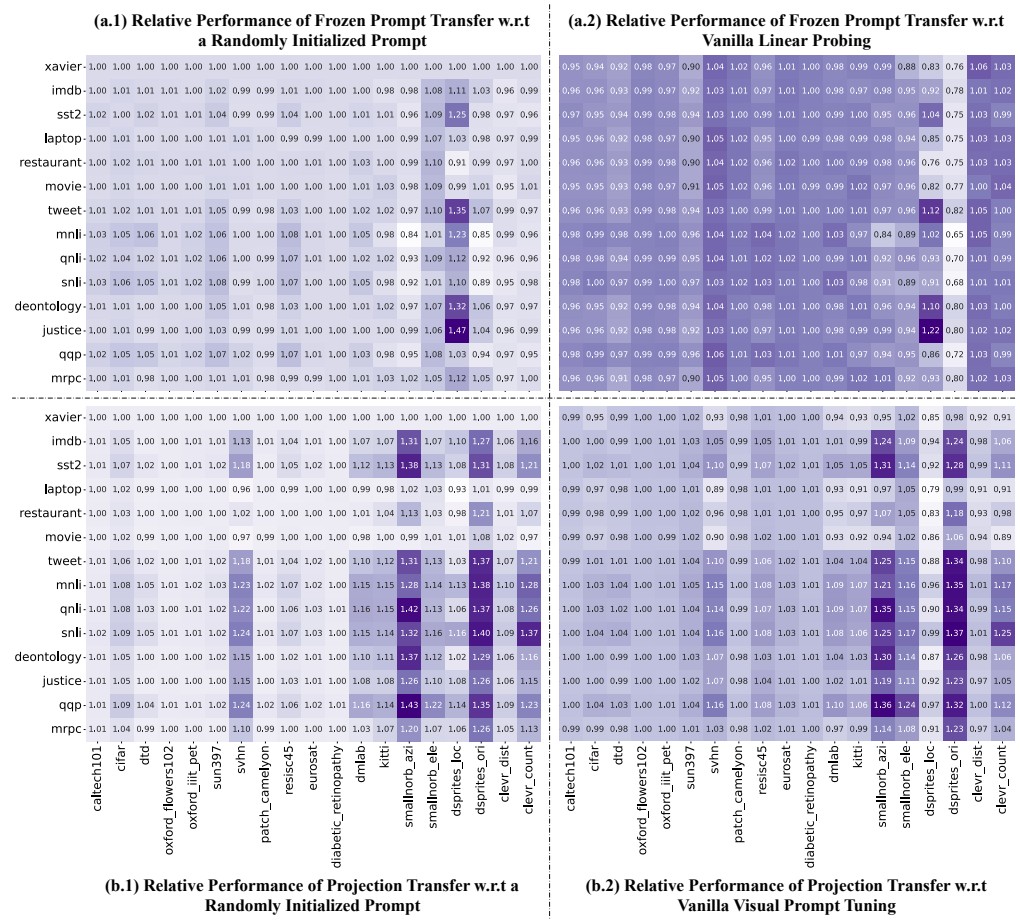

Figure 3: **(a) Relative Performance of Frozen Prompt Transfer** concerning (1) a random prompt, and (2) vanilla linear probing. **(b) Relative Performance of Projection Transfer** concerning (1) a random prompt, and (2) vanilla VPT. The vertical axis shows the NLP tasks on which the source prompts are trained, while the horizontal shows the CV tasks the source prompts are transferred to. `xavier` means the prompt is not trained on an NLP task but randomly initialized using Xavier initialization (Glorot & Bengio, 2010) and also kept frozen during transfer. The relative performance is calculated by $(\text{Performance} - B)/B \times 100\%$, where $B$ is the performance of transferring a random prompt, conducting vanilla linear probing or VPT, depending on the situations stated in the figure.

$\{0, 0.0001, 0.001, 0.01\}$, with a batch size of 64, warm-up steps of 10, cosine learning rate scheduler, and an Adam optimizer. The complete set of hyperparameters of different scenarios on each CV task is listed in the Appendix.

## 4.2 CROSS-MODALITY PROMPT TRANSFER

In this section, frozen prompt transfer is first carried out to verify the feasibility of transferring prompts across modalities. The results are reported in Figure 3a, where the relative performance compared with two different baselines is reported: (i) frozen prompt transfer using a randomly initialized prompt, and (ii) vanilla linear probing (re-ran by us).

Later on, to adapt the linguistic knowledge stored in the source prompts to CV tasks and study the cross-modality prompt transferability, projection transfer is carried out, with the results reported in Figure 3b. Similarly, the performance is reported in two different formats: (i) the relative performance concerning projection transfer with a randomly initialized prompt. (ii) the relative performance pertaining to vanilla VPT (with a fixed prompt length of 100, re-ran by us). The exact accuracy of linear probing, VPT, and projection transfer (only the accuracy with the best source prompt) is

reported in Table 1. All other accuracy results are left in the Appendix. Based on Figure 3 and Table 1, the following key conclusions are drawn:

**(i) Source prompts pretrained on text data can be safely transferred to tasks in a different modality.** This claim is supported by the fact that the trained prompts can help a pretrained ViT achieve better linear probing performance compared to a random prompt and vanilla linear probing (Figure 3a), inferring that the feature clusters extracted by the ViT model will benefit from simply prepending the frozen source prompts to the image embeddings. When compared with a random prompt, 150 out of 247 transfer pairs ($61.5\%$) achieve better performance. This number decreases when compared with vanilla linear probing, but still holds a value of 78 ($31.6\%$), demonstrating the potential for transferring NLP prompts across modalities.

**(ii) The linguistic knowledge stored in the source prompt is helpful on a different modality task after adaptation.** In Figure 3b.1, $86.2\%$ of the transfer pairs (213 out of 247) can achieve better performance compared to conducting projection transfer using a random prompt. This demonstrates the importance of the linguistic knowledge stored in the source prompts, which is obtained through pretraining the source prompts on NLP tasks. In Figure 3b.2, $64\%$ of the transfer pairs (158 out of 247) are better than vanilla VPT, with the largest boost observed on SNLI-to-Dsprites/loc. This demonstrates that the linguistic knowledge learned by the source prompts can be of great help on a different modality task after adaptation. Additionally, the improvements observed in Figure 3b.2 are proven to be statistically significant: they are caused by transferring the source prompts, instead of error or random chances (the prove can be found in our appendix).

**(iii) More source data should yield better transfer performance, but this is not always the case.** Normally, in transfer learning, more source data would bring better performance on the target task. This pattern also holds on projection transfer: performant source prompts (best first: SNLI, MNLI, QQP, and QNLI) tend to have been pretrained on a vast amount of text data: 549K sentence pairs for SNLI; 392k sentence pairs for MNLI; 363k sentence pairs for QQP; and 104k sentence pairs for QNLI. The opposite also runs true: the worst source prompts (laptop, restaurant, and movie) are only pretrained on less than 3k sentences. This general pattern holds basically for every target CV task. However, the volume of pretraining data does not solely determine the transferability of the source prompts, as plenty of counterexamples can be found in Figure 3b (SNLI does not always achieve the best performance on the target CV task). To this end, we take a different perspective that measures both the modality and task gaps as the cross-modality prompt transferability indicator.

**(iv) Prompt tuning performance on target tasks that already benefit a lot from prompt tuning is likely to be further boosted by cross-modality prompt transfer.** This claim is drawn from the relation between rows *V/L* and *P/V* in Table 1. V/L shows the relative performance gain of VPT over linear probing (V/L gain). P/V shows the relative performance gain of projection transfer over VPT (P/V gain). To visualize the relation more intuitively, we plot the values of V/L gain and P/V gain in Figure 4, which shows a clear positive relation between V/L gain and P/V gain. Looking deeper into the values in Table 1: tasks with a V/L gain lower than 3% have an average P/V gain of only 1.7%, while those with a V/L gain greater than 3% come with an average P/V gain of 14.1%. Although counterexamples or outliers (such as dSprites/loc) can be found,

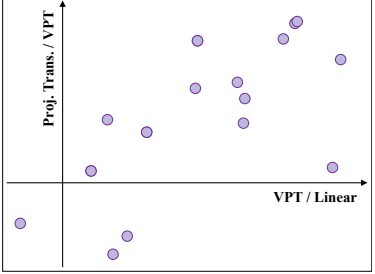

Figure 4: Scatter plot of V/L gain vs P/V gain. The values are log-scaled for better visualization.

the overall trend still informs us that, if a task benefits a lot from prompt tuning, then its prompt tuning performance is likely to be further boosted by cross-modality prompt transfer.

### 4.3 PROMPT TRANSFERABILITY ESTIMATION

For a target CV task, we obtain the transferability scores for each source prompt by two steps: (i) Train the target prompt (with random initialization) on the target task using vanilla VPT and (ii) Perform Equation 6 to obtain the transferability score for each source prompt. Note: when training the universal projector for measuring the task gap, a learning rate of 1e-4, weight decay of 0.1, batch size of 13 (all source prompts are loaded in the same batch), and the Adam optimizer are used. This set of hyperparameters is grid-searched on CIFAR100 and extended to every target task. The universal

Table 1: Accuracy of linear probing (*Linear*), visual prompt tuning (*VPT*), and projection transfer (*Proj.*). $V/L = (VPT - Linear)/Linear \times 100\%$ shows the relative gain of *VPT* over *Linear*. $P/V = (Proj. - VPT)/VPT \times 100\%$ shows the relative gain of *Proj.* over *VPT*. Note: (i) The *VPT* re-ran by us uses a uniform prompt length of 100 on all tasks. (ii) For *Proj.*, only the result obtained by the best source prompt is reported.

| | Caltech101 | CIFAR100 | DTD | Flowers102 | Pets | Sun397 | SVHN | Patch Camelyon | Resisc45 | EuroSAT | Retinopathy | DMLab | KITTI | SmallNORB/azi | SmallNORB/ele | dSprites/loc | dSprites/ori | Clevr/dist | Clevr/count |
|---|---|---|---|---|---|---|---|---|---|---|---|---|---|---|---|---|---|---|---|
| Linear | 86.3 | 65.3 | 64.4 | 97.3 | 86.6 | 51.2 | 36.3 | 79.0 | 69.5 | 89.3 | 74.3 | 33.8 | 60.1 | 11.9 | 20.9 | 12.7 | 21.0 | 31.2 | 36.3 |
| VPT | 88.7 | 78.0 | 65.7 | 97.8 | 87.9 | 50.2 | 71.3 | 80.8 | 75.7 | 92.9 | 74.0 | 39.8 | 72.2 | 17.4 | 22.8 | 72.9 | 30.9 | 57.6 | 50.0 |
| V/L | +2.9% | +19.6% | +2.1% | +0.5% | +1.6% | −2.0% | +96.6% | +2.3% | +8.9% | +4.0% | −0.5% | +17.7% | +20.0% | +45.6% | +9.2% | +475.7% | +47.4% | +84.5% | +37.7% |
| Proj. | 89.0 | 81.0 | 68.4 | 98.2 | 89.0 | 52.6 | 82.6 | 81.0 | 82.0 | 95.8 | 74.8 | 43.6 | 77.0 | 23.6 | 28.3 | 72.5 | 42.4 | 58.4 | 62.6 |
| P/V | +0.3% | +3.8% | +4.1% | +0.4% | +1.3% | +4.8% | +15.8% | +0.2% | +8.3% | +3.1% | +1.1% | +9.5% | +6.6% | +35.6% | +24.1% | −0.5% | +37.2% | +1.4% | +25.2% |

projector is trained for 100 epochs and only the last-epoch universal projector is used. More training details can be found in the Appendix.

After the transferability scores of all the source prompts are obtained for a target task, the scores are ranked from highest to lowest. The ranking results are compared with the ranked projection transfer results on the target task with Kendall's coefficient (Kendall, 1938) as the evaluation metric. Kendall's coefficient is often used to determine the strength and direction of the relationship between two ranked variables. It ranges from -1 to 1. Greater values indicate better transferability estimation results. The Kendall's coefficient for the investigated target tasks is listed in Table 2, which involves the following methods:

- Average cosine similarity (**Avg Cos**) that average-pools the source prompt vectors into one vector and calculates the cosine similarity between the average-pooled source and target prompts. This method is investigated by Vu et al. (2022) and Su et al. (2022).

- Model stimulation similarity (**ON**, introduced by Su et al. (2022)) that feeds the source and target prompts to the language model separately and calculates the overlapping rate between the attention block's activation patterns of the source and target prompts. In our case, since the source prompt will become part of ViT's input, we feed the source and target prompts to ViT and obtain the activation map of the last three attention layers of ViT to calculate the overlapping rate, given that Su et al. (2022) had verified that using the last three layers would give the best result.

- $V_{data}$ that ranks the source prompts according to the data amount they were pretrained on. This simple method is involved as a support for a former claim that "*more source data would yield better transfer performance*", showing the high relevancy between the amount of pretraining data and cross-modality prompt transferability. Despite the high relevance, $V_{data}$ cannot serve as a reliable prompt transferability estimation method, as the ranking results are determined purely by the attributes of source prompts, leaving no target information taken into consideration. Therefore, $V_{data}$ will not be involved in the comparison.

- $\mathcal{G}_M$ **Only** and $\mathcal{G}_T$ **Only** use the modality gap $\mathcal{G}_M$ or task gap $\mathcal{G}_T$ as the transferability metric. We apply these two baselines to verify our hypothesis and show the importance of estimating the modality gap and task gap concurrently.

From the results in Table 2, it is clear that by measuring both the modality and task gaps at the same time the cross-modality prompt transferability can be well estimated. Looking deeper, we can observe that measuring only $\mathcal{G}_M$ can obtain a result much better than measuring only $\mathcal{G}_T$ and is very close to $\mathcal{G}_M$ & $\mathcal{G}_T$. We hypothesize that in cross-modality scenarios, the modality gap would normally be more influential to the transfer performance than the task gap. Moreover, it's interesting to note that the only difference between *Avg Cos* and $\mathcal{G}_T$ *Only* is the involvement of the universal projector, without which the cosine similarity cannot serve as a solid approach for estimating cross-modality prompt transferability. This verifies that source and target prompts lie in different semantic spaces and direct similarity calculation will not be meaningful. For the baseline method ON, it achieves relatively good results on most of the target tasks, even if the source and target prompts lie in different semantic spaces, demonstrating that it's a promising direction for diving deeper in the future.

Table 2: Kendall's coefficient (scaled to $-100 \sim 100$) calculated on each target task by different prompt transferability estimation methods. Combining the modality and task gaps yields the best transferability estimation result. The best results are **bold** while the best but equal results are underlined ($V_{data}$ that ranks the source prompts based on pretrained data volumes will not be included for comparison). The exact ranking results can be found in the Appendix.

| | Caltech101 | CIFAR100 | DTD | Flowers102 | Pets | Sun397 | SVHN | Patch Camelyon | Resisc45 | EuroSAT | Retinopathy | DMLab | KITTI | SmallNORB/azi | SmallNORB/ele | dSprites/loc | dSprites/ori | Clevr/dist | Clevr/count | Mean |
|---|---|---|---|---|---|---|---|---|---|---|---|---|---|---|---|---|---|---|---|---|
| Avg Cos | 1.28 | -33.33 | 12.82 | 19.23 | 0.00 | 6.41 | -11.54 | 0.00 | 10.26 | -30.77 | -32.05 | -51.28 | -66.67 | 46.15 | -23.08 | -20.51 | -28.21 | -35.90 | 30.77 | -10.86 |
| ON | 6.41 | 61.54 | 7.69 | 62.82 | 17.95 | 21.79 | 24.36 | 23.08 | -58.97 | -53.85 | -39.74 | 2.56 | 47.44 | 0.00 | 46.15 | 12.82 | -17.95 | 58.97 | 46.15 | 14.17 |
| $V_{data}$ | 73.08 | 87.18 | 82.05 | 80.77 | 64.10 | 70.51 | 85.90 | 71.79 | 92.31 | 87.18 | 55.13 | 76.92 | 71.79 | 64.10 | 84.62 | 64.10 | 82.05 | 87.18 | 89.74 | 77.39 |
| $\mathcal{G}_M$ Only | 75.64 | 92.31 | **89.74** | **78.21** | 64.10 | 78.21 | 80.77 | **74.36** | 87.18 | 79.49 | **65.38** | 89.74 | 84.62 | 66.67 | 87.18 | 56.41 | 84.62 | 89.74 | 92.31 | 79.83 |
| $\mathcal{G}_T$ Only | 62.82 | 79.49 | 71.79 | 65.38 | 64.10 | 70.51 | 78.21 | 61.54 | 71.79 | 89.74 | 57.69 | 87.18 | 79.49 | 61.54 | 87.18 | 53.85 | 82.05 | 82.05 | 71.79 | 72.54 |
| $\mathcal{G}_M$ & $\mathcal{G}_T$ | **78.21** | **94.87** | 87.18 | 75.64 | **66.67** | 75.64 | **83.33** | 69.23 | 87.18 | 89.74 | 62.82 | 89.74 | 84.62 | **69.23** | **89.74** | **58.97** | 84.62 | 89.74 | **94.87** | **80.63** |

Table 3: Accuracy of Self-Prompt Tuning (*SPT*) and attention transfer (*Attn.*). Note: (i) *: the accuracy of the baseline SPT is obtained from its original paper. (ii) The best accuracy is **bolded**.

| | Natural | | | | | | | | Specialized | | | | | Structured | | | | | | | | |
|---|---|---|---|---|---|---|---|---|---|---|---|---|---|---|---|---|---|---|---|---|---|---|
| | Caltech101 | CIFAR100 | DTD | Flowers102 | Pets | Sun397 | SVHN | Mean. | Patch Camelyon | Resisc45 | EuroSAT | Retinopathy | Mean. | DMLab | KITTI | SmallNORB/azi | SmallNORB/ele | dSprites/loc | dSprites/ori | Clevr/dist | Clevr/count | Mean. |
| SPT* | **91.2** | 78.9 | **71.2** | **99.4** | **90.7** | 52.3 | **86.4** | **81.44** | **82.5** | 82.6 | 94.9 | 74.6 | 83.65 | **48.0** | 68.8 | **24.4** | **37.4** | 72.6 | 41.9 | **61.2** | **68.6** | 52.86 |
| Attn. | 90.1 | **81.7** | 68.9 | 98.4 | 89.4 | **52.6** | 82.8 | 80.55 | 82.1 | **82.9** | **96.2** | **75.3** | **84.12** | 44.9 | **77.4** | 24.3 | 36.3 | **73.1** | **43.2** | 60.8 | 67.9 | **53.47** |

## 4.4 ATTENTION TRANSFER

To further demonstrate the effectiveness of cross-modality prompt transfer, attention transfer is compared with the newest prompt-based benchmark: Self-Prompt Tuning (SPT, Wang et al. (2024b)). SPT focuses on prompt initialization for boosting prompt tuning: it initializes the prompts with sampled image patch embeddings, which can be regarded as injecting prior target data knowledge into the prompt. Similarly, attention transfer can also be regarded as injecting knowledge into the prompt: the final prompt of attention transfer contains knowledge not only from the source NLP task but also from the target CV task. Due to this similarity and the timeliness, SPT is chosen as the comparison baseline. The results are reported in Table 3.

From the results: attention transfer can achieve a comparable average performance on *Natural* tasks and better average performance on *Specialized* and *Structured* tasks. This serves as strong evidence for cross-modality prompt transfer being able to boost prompt tuning. Note that SPT is not a baseline focusing on prompt transfer. Instead, it focuses on the prompt initialization strategy. Yet, attention transfer largely depends on the source prompt, whose ability on the target task is further restricted by the modality and task gaps, which, do not appear in SPT. Therefore, attention transfer functions under a more challenging setup, but still achieves an average performance comparable to or even better than SPT. Given that SPT and attention transfer can be regarded as two different perspectives for prompt initialization, even though attention transfer falls below SPT on some tasks, the results still demonstrate the potential of cross-modality prompt transfer.

## 5 CONCLUSIONS

In this paper, we conducted extensive experiments to verify the feasibility and study the prompt transferability for cross-modality prompt transfer. We also proposed a prompt transferability estimation method based on gap analysis. Lastly, we further demonstrated the potential and effectiveness of cross-modality prompt transfer through our proposed attention transfer, boosting prompt tuning to a level that matches and even outperforms the newest prompt-based PETL benchmark. Our research opened a new path to benefiting data-scarce tasks with the rich resources of the text modality. We hope our research can serve as a solid foundation for the topic of cross-modality prompt transfer and provide new directions and inspirations to mitigating challenges faced by data-scare tasks.

ACKNOWLEDGMENT

The work was supported by the Australian Research Council (ARC) under Laureate Fellow grant FL190100149.

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

# APPENDIX

# Contents

# A INFLUENTIAL PRIOR STUDIES

Our work is built upon a series of prior studies, which offered great help to the conceptualization, implementation, and conclusion of our work. We list these influential prior studies in Table 1, along with their contributions to our study. Here we wish to express our sincere thanks to the authors for their great and reproducible works, without which there will be no begins of our work.

Table 1: Influential prior studies that are of great help to our work.

| Title & Author(s) | Venue | How they help our research |
|---|---|---|
| Visual Prompt Tuning (Jia et al., 2022) | ECCV2022 | The conceptualization of VPT & the publicly available training receipes |
| Sensitivity-Aware Visual Parameter-Efficient Tuning (He et al., 2023) | ICCV2023 | The publicly available dataset processing script of VTAB-1K |
| SPoT: Better Frozen Model Adaptation through Soft Prompt Transfer (Vu et al., 2022) | ACL2022 | The conceptualization of prompt transfer |
| On Transferability of Prompt Tuning for Natural Language Processing (Su et al., 2022) | NAACL2022 | The publicly available codes and trained prompts |

# B PRELIMINARIES

## B.1 MAXIMUM MEAN DISCREPANCY

Maximum Mean Discrepancy (MMD) is a non-parametric method used to measure the difference between two probability distributions $P$ and $Q$ based on samples drawn from these distributions. The MMD is particularly useful in two-sample hypothesis testing to determine whether $P = Q$.

Given two sets of samples $\{x_1, \ldots, x_m\}$ drawn from distribution $P$ and $\{y_1, \ldots, y_n\}$ drawn from distribution $Q$, MMD is defined as the distance between the mean embeddings of the two distributions in a reproducing kernel Hilbert space (RKHS). Mathematically, the MMD is computed as:

$$\text{MMD}^2(P, Q; \mathcal{H}) = \|\mathbb{E}_P[\phi(x)] - \mathbb{E}_Q[\phi(y)]\|_{\mathcal{H}}^2 \tag{1}$$

where $\phi$ is a feature map into the RKHS $\mathcal{H}$. Expanding the square norm gives the following expression:

$$\text{MMD}^2(P, Q; \mathcal{H}) = \mathbb{E}_{x,x' \sim P}[k(x, x')] + \mathbb{E}_{y,y' \sim Q}[k(y, y')] - 2\mathbb{E}_{x \sim P, y \sim Q}[k(x, y)] \tag{2}$$

Here, $k(x, y)$ is a positive-definite kernel function, such as the Gaussian or linear kernel, which defines the inner product in the RKHS.

Intuitively, the MMD measures how well the mean of the samples from $P$ and $Q$ match in the RKHS. If the distributions are identical, the MMD will be zero.

## B.2 PROMPT TRANSFERABILITY ESTIMATION: MODEL STIMULATION SIMILARITY

The prompt transferability estimation method proposed by Su et al. (2022) is chosen as our baseline. However, due to page limitation, their method (named ON) is not introduced in detail. Therefore in this section, we provide a brief but detailed introduction on their proposed method.

On measures the prompt transferability between two NLP tasks $s_1$ and $s_2$ by the following steps: (i) feed the prompt $p_{s_1}$ (trained on $s_1$) solely to RoBERTa and record its activation map $A_1$. The activation map of a single feed-forward layer is a binary vector obtained by setting the activation values greater than zero to 1 and 0 otherwise. The final activation map is the concatenation of the single-layer activation maps of the last three attention blocks. (ii) perform step (i) on the prompt $p_{s_2}$ (trained on $s_2$) to obtain its activation map $A_2$. (iii) calculate the cosine similarity between $A_1$ and $A_2$ as the final transferability score between $s_1$ and $s_2$:

$$\text{ON}(p_{s_1}, p_{s_2}) = \frac{A_1 \times A_2}{||A_1|| \times ||A_2||} \tag{3}$$

In our re-implementation, since the source prompts are transferred to CV tasks and will be used on a pretrained ViT model, we use ViT instead of RoBERTa to calculate the model stimulation similarity between a source prompt and a target prompt trained on a CV task.

# C  TECHNICAL DETAILS

## C.1  IMAGE PROCESSING AND AUGMENTATION

For the VTAB-1K (Zhai et al., 2019) dataset, we use the scripts provided by He et al. (2023)[1] to download and convert the dataset into PNG images. To load the images into our codes, we use the `Dataset` class provided by PyTorch (Paszke et al., 2019).

Regarding the image augmentation strategies for VTAB-1K, we follow the default settings in VTAB-1K and do not use any augmentation tricks except the following:

1. Resizing the images to a size of $224 \times 224$;

2. Converting the images to PyTorch tensors and re-scaling them to $0 \sim 1$;

3. Normalizing the images using the mean and standard deviation values calculated from ImageNet ($mean = (0.485, 0.456, 0.406)$, $std = (0.229, 0.224, 0.225)$).

## C.2  TRAINING PROMPTS ON NLP TASKS

The source prompts trained with RoBERTa on different NLP tasks are directly adopted from the official code repository of Su et al. (2022)[2]. Below we show some crucial information for training the source prompts:

**Input Embedding Structures.**  For NLP tasks that input one sentence at a time (such as sentiment analysis), the input embedding structure in Table 2 is adopted. While for NLP tasks that input two sentences at the same time (such as natural language inference, ethical judgment, and paraphrase identification), the input embedding structure in Table 3 is adopted.

In the tables, the top row shows the order of different tokens. The *Length* row indicates the length of each token. The *Learnable?* row indicates whether the corresponding tokens are learnable or not. Finally, the *Positional?* row indicates whether the positional embeddings of the pretrained RoBERTa are added to the corresponding token.

Table 2: The input embedding structure for training prompts on single-sentence NLP tasks.

|  | [MASK] | [Prompt] | [CLS] | [Input Sentence] | [SEP] |
|---|---|---|---|---|---|
| Length | 1 | 99 | 1 |  | 1 |
| Learnable? | √ | √ | × | × | × |
| Positional? | × | × | √ | √ | √ |

Table 3: The input embedding structure for training prompts on dual-sentence NLP tasks.

|  | [MASK] | [Prompt] | [CLS] | [Input Sentence 1] | [SEP] | [Input Sentence 2] | [SEP] |
|---|---|---|---|---|---|---|---|
| Length | 1 | 99 | 1 |  | 1 |  | 1 |
| Learnable? | √ | √ | × | × | × | × | × |
| Positional? | × | × | √ | √ | √ | √ | √ |

**Hyperparameters.**  In fact, the hyperparameters used to train the prompts on different NLP tasks will be omitted here. The complete training recipes on each NLP task, including the hyperparameters, optimizer, learning rate scheduler, and so on, can be found in the original code repository of Su et al. (2022). This link shall direct you to their training recipes.

---

[1] https://github.com/ziplab/SPT
[2] https://github.com/thunlp/Prompt-Transferability

## C.3 Transferring Prompts to CV Tasks

**Input Embedding Structures.** In all of our transfer settings, as long as there is prompt prepended to the image patch embeddings, no matter how the prompt is obtained, the prompt length will always be 100 and the input embedding structure demonstrated in Table 4 is adopted. This applies to not only cross-modality prompt transfer scenarios, but also to vanilla visual prompt tuning scenarios.

Table 4: The input embedding structure for ViT.

|            | [Prompt]   | [CLS]      | [Image Patch Embedding] |
|------------|------------|------------|-------------------------|
| Length     | 100        | 1          | 196                     |
| Learnable? | $\checkmark$ | $\times$ | $\times$                |
| Positional? | $\times$  | $\checkmark$ | $\checkmark$          |

**Frozen Prompt Transfer.** Frozen Prompt Transfer simply prepends the trained but frozen source prompts to the image patch embeddings and only learns an output predictor on top of the [CLS] features. The only thing that needs attention is the output predictor: it consists of a linear layer (with bias) whose input dimension is the hidden dimension of ViT (768 in our case) and output dimension is the number of classes of the target CV task. The same output predictor is used across every settings in our paper.

**Projection Transfer.** The core module in projection transfer is the linear projector, which is a simple linear layer (with bias) with an input dimension of 768 (hidden dimension of RoBERTa) and output dimension of 768 (hidden dimension of ViT).

## C.4 Cross-Modality Prompt Transferability Estimation

### C.4.1 Modality Gap

In this paper, the modality gap between a source NLP task and a target CV task is measured as the Maximum Mean Discrepancy (MMD, Gretton et al. (2012)) between source prompts and target image patch embeddings. This section will explain on how to calculate the MMD between a source prompt (100 vectors, each with a dimension of 768) and the target image patch embeddings ($1000 \times 196$ vectors, each with a dimension of 768). Note that we use the training set of each target task for MMD calculation. This means that for every target task, we only use 1000 images for MMD calculation. Since an image will be converted into 196 patch embeddings, we would have $1000 \times 196$ vectors in each target CV task for MMD calculation. The procedure of measuring the modality gap is illustrated in Algorithm 1.

---

**Algorithm 1** Measuring modality gap via MMD

---

**Input:** Source prompt $p_s$ with dimension $(l, d_{hs})$; Vision task $\mathcal{T}_t$; ViT's image embedder $E_v$
**Output:** Measured modalit gap between $p_s$ and $\mathcal{T}_t$
1: Extract patch embeddings: $e_i = E_v(\mathcal{T}_t)$ (dimension: $(|\mathcal{T}_t|, N_p, d_{hs})$, $N_p$ is the # of patches)
2: Reshape $e_i$ to $(|\mathcal{T}_t| \times N_p, d_{hs})$; Then shuffle $e_i$ along the 0-th axis
3: $(M_{total}, N_{steps}) = (0, 0)$
4: **for** *epoch* in *range(5)* **do**
5:     **while** $e_i$ is not sampled out **do**
6:         Sample $l$ embeddings from $e_i$, denoted as $\hat{e}_i$
7:         Calculate the MMD between $p_s$ and $\hat{e}_i$: $M_{current} = MMD(p_s, \hat{e}_i)$
8:         $M_{total} += M_{current}$
9:         $N_{steps} += 1$
10:     **end while**
11: **end for**
12: **return** $M_{total}/N_{steps}$

---

## C.4.2 TASK GAP

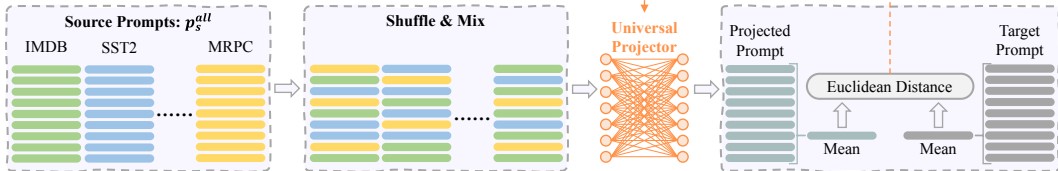

Figure 1: Training the universal projector for estimating the task gap.

In this paper, the task gap between a source NLP task and a target CV task is measured as the cosine similarity between the projected source prompt and the target prompt obtained through vanilla visual prompt tuning on the target CV task. The core module here is the universal projector that projects all the source prompts to the space of the target prompt. The universal projector itself is a simple linear projector (with bias) with an input dimension of 768 (the hidden dimension of RoBERTa) and an output dimension of 768 (the hidden dimension of ViT). Given a target CV task, Figure 1 demonstrates the procedure for training the universal prompt projector:

**Step 0** Perform vanilla visual prompt tuning on the target CV task with a pretrained ViT to obtain the target prompt $p_t$.

**Step 1** Shuffle all the source prompts trained on each NLP task and mix them with each other. In total there are 13 source tasks, which means that if we gather all the source prompts, we would have $13 \times 100$ vectors, each with a dimension of 768. To shuffle and mix them, we simply rearrange the $(13, 100, 768)$ vectors into $(1300, 768)$ vectors, shuffle them along the 0-th axis and rearrange them back to the shape of $(13, 100, 768)$.

**Step 2** Pass the shuffled-and-mixed source prompts to the universal projector $P_u$. Let's denote the projected source prompts as $P_u(p_s^{all})$, the universal projector $P_u$ is updated by the Euclidean distance between $P_u(p_s^{all})$ and $p_t$:

$$\arg\min_{P_u} ||mean(P_u(p_s^{all})) - mean(p_t)||_2 \tag{4}$$

In practical, $mean(P_u(p_s^{all}))$ has a shape of $(13, 768)$ while $mean(p_t)$ has a shape of $(1, 768)$, the final Euclidean distance is calculated by averaging the total Euclidean distance obtained by summing up the distance between $mean(p_t)$ and the 13 vectors from $mean(P_u(p_s^{all}))$.

After the universal projector is trained, the task gap between a source task and the target CV task can be measured by the cosine similarity between the projected source prompt and the target prompt. The reason why the source prompts are shuffled and mixed is two-fold: (i) To avoid overfitting and reduce susceptibility on hyperparameters, as in practical, we will not have ground-truth results to tune the hyperparameters. (ii) To further boost the prompt transferability estimation performance. The effects of the shuffling and mixing operations will be demonstrated in Section C.5.4.

## C.5 PER-TASK TRAINING HYPER-PARAMETERS

### C.5.1 LINEAR PROBING

The complete set of hyperparameters for linear probing with respect to each target task can be viewed in Table 5. The total number of epochs is 100. Note: (i) For linear probing, we did not perform hyper-parameter grid-search on any of the target tasks, the hyper-parameters in the table is released by Jia et al. (2022) and we simply follow. (ii) The *Base LR* column indicates the base learning rate for each target task. This base learning rate will be scaled according to batch size: Final LR = (Base LR ∗ batch size)/256, before the training starts. (iii) The linear probing performance reported in our main paper is obtained by our re-implementation, with is nearly identical to that reported by Jia et al. (2022). Therefore, we did not question this set of hyper-parameters at all. (iv) The set of hyper-parameters is also applied to frozen prompt transfer, regardless of the source

prompts used. It would be highly unrealistic for us to perform grid-search on frozen prompt transfer whenever the source prompt is changed.

Table 5: Hyper-parameters for linear probing and frozen prompt transfer.

| Task | Optimizer | Momentum | Base LR | LR Warm-Up | LR Decay | Weight Decay | Batch Size |
|------|-----------|----------|---------|------------|----------|--------------|------------|
| Caltech101 | SGD | 0.9 | 2.5 | 10 | Cosine | 0.001 | 2048 |
| CIFAR100 | SGD | 0.9 | 0.1 | 10 | Cosine | 0 | 2048 |
| DTD | SGD | 0.9 | 1 | 10 | Cosine | 0 | 2048 |
| Flowers102 | SGD | 0.9 | 0.1 | 10 | Cosine | 0.001 | 2048 |
| Pets | SGD | 0.9 | 0.25 | 10 | Cosine | 0.001 | 2048 |
| Sun397 | SGD | 0.9 | 0.5 | 10 | Cosine | 0 | 2048 |
| SVHN | SGD | 0.9 | 0.25 | 10 | Cosine | 0.01 | 2048 |
| Patch Camelyon | SGD | 0.9 | 0.05 | 10 | Cosine | 0.01 | 2048 |
| Resisc45 | SGD | 0.9 | 0.5 | 10 | Cosine | 0 | 2048 |
| EuroSat | SGD | 0.9 | 10 | 10 | Cosine | 0.0001 | 2048 |
| Retinopathy | SGD | 0.9 | 0.1 | 10 | Cosine | 0.01 | 2048 |
| DMLab | SGD | 0.9 | 0.5 | 10 | Cosine | 0 | 2048 |
| Kitti | SGD | 0.9 | 5 | 10 | Cosine | 0 | 2048 |
| SmallNORB/azi | SGD | 0.9 | 2.5 | 10 | Cosine | 0.01 | 2048 |
| SmallNORB/ele | SGD | 0.9 | 2.5 | 10 | Cosine | 0.01 | 2048 |
| dSprites/loc | SGD | 0.9 | 5 | 10 | Cosine | 0 | 2048 |
| dSprites/ori | SGD | 0.9 | 0.1 | 10 | Cosine | 0 | 2048 |
| Clevr/dist | SGD | 0.9 | 0.5 | 10 | Cosine | 0.001 | 2048 |
| Clevr/count | SGD | 0.9 | 0.1 | 10 | Cosine | 0 | 2048 |

### C.5.2 VANILLA VISUAL PROMPT TUNING

The complete set of hyperparameters for vanilla visual prompt tuning with respect to each target task can be viewed in Table 6. The total number of epochs is 100. Note: (i) The original VPT repository also has a set of carefully grid-searched hyperparameters for VPT. However, the original VPT adopt different prompt lengths on different target CV tasks, while in our experiments we fix the prompt length to 100 on every target task. Therefore, their hyperparameters may not be suitable for our re-implementation. The hyperparameters in the table are obtained by our grid-search results. (ii) However, the original VPT uses a prompt length of 100 on the following tasks: CIFAR100, DMLab, Kitti, DSprites/loc, dSprites/ori, and Clevr/count. For these tasks, we also perform grid search to find a presumably best combination of base learning rate and weight decay values. We later perform VPT using this set of values and compare the test accuracy with that obtained by the original set of hyperparameters. Finally, we take the set of hyperparameters that gives the best test accuracy. Among the tasks with a prompt length of 100, CIFAR100, DMLab, Kitti, and dSprites/ori keep using the original set of hyperparameters. While DSprites/loc and Clevr/count use the set of hyperparameters grid-searched by us. (iii) Similar to linear probing, the base learning rate will be scaled according to batch size: Final LR $= ($Base LR $*$ batch size$)/256$, before the training starts.

### C.5.3 CROSS-MODALITY PROMPT TRANSFER

For frozen prompt transfer, the set of hyperparameters is exactly the same to linear probing regardless of the source prompt prepended, as depicted in Table 5. The total number of epochs is also 100. It is worth mentioning that for some of the target tasks (particularly SmallNORB/azi, SmallNORB/ele, dSprites/loc, and dSprites/ori), we observed extremely big loss values ($> 1000$) when using some specific source prompts (particularly MNLI, QNLI, SNLI, and QQP prompts that are performant in projection transfer). We suspect that the hyperparameters used in linear probing is not the optimal hyperparameters for these transfer tasks. However, as mentioned before, its impossible for us the search for the best set of hyperparameters for every source-target pair, given the fact that there are 247 pairs in total. What we can hope is an universal set of hyperparameters that can highlight the relative competency of the source prompts. Therefore, we did not try to draw any other conclusions but "*whether prompts trained on NLP tasks can be safely transferred to CV tasks*" from the experimental results of frozen prompt transfer.

For projection transfer, the Adam (Kingma & Ba, 2014) optimizer is used, with a learning rate of 0.005, a weight decay of 0.001, and a batch size of 64. No warm up steps or learning rate scheduler are adopted. The total number of training epochs is also 100. This set of hyperparameters is grid-searched

Table 6: Hyper-parameters for vanilla visual prompt tuning.

| Task | Optimizer | Momentum | Base LR | LR Warm-Up | LR Decay | Weight Decay | Batch Size |
|---|---|---|---|---|---|---|---|
| Caltech101 | SGD | 0.9 | 10 | 10 | Cosine | 0.001 | 64 |
| CIFAR100 | SGD | 0.9 | 10 | 10 | Cosine | 0.001 | 64 |
| DTD | SGD | 0.9 | 5 | 10 | Cosine | 0.001 | 64 |
| Flowers102 | SGD | 0.9 | 2.5 | 10 | Cosine | 0.001 | 64 |
| Pets | SGD | 0.9 | 2.5 | 10 | Cosine | 0.001 | 64 |
| Sun397 | SGD | 0.9 | 10 | 10 | Cosine | 0.001 | 64 |
| SVHN | SGD | 0.9 | 0.5 | 10 | Cosine | 0.01 | 64 |
| Patch Camelyon | SGD | 0.9 | 0.5 | 10 | Cosine | 0.01 | 64 |
| Resisc45 | SGD | 0.9 | 10 | 10 | Cosine | 0.001 | 64 |
| EuroSat | SGD | 0.9 | 5 | 10 | Cosine | 0.0001 | 64 |
| Retinopathy | SGD | 0.9 | 2.5 | 10 | Cosine | 0.01 | 64 |
| DMLab | SGD | 0.9 | 500 | 10 | Cosine | 0 | 64 |
| Kitti | SGD | 0.9 | 250 | 10 | Cosine | 0 | 64 |
| SmallNORB/azi | SGD | 0.9 | 10 | 10 | Cosine | 0.001 | 64 |
| SmallNORB/ele | SGD | 0.9 | 10 | 10 | Cosine | 0.001 | 64 |
| dSprites/loc | SGD | 0.9 | 0.5 | 10 | Cosine | 0.01 | 64 |
| dSprites/ori | SGD | 0.9 | 0.5 | 10 | Cosine | 0.01 | 64 |
| Clevr/dist | SGD | 0.9 | 0.1 | 10 | Cosine | 0.01 | 64 |
| Clevr/count | SGD | 0.9 | 0.5 | 10 | Cosine | 0.01 | 64 |

on some of the target tasks with a source prompt trained on IMDB. We found that multiple target tasks give similar grid-search results. Therefore we ceased searching on the rest of the tasks and extend the set of hyperparameters to every source-target pair.

### C.5.4 TRAINING THE UNIVERSAL PROJECTOR FOR MEASURING TASK GAP

Given a target CV task, to measure the task gap between it and all the source NLP tasks, the universal projector needs to be trained. The function of the universal projector is to project all the source prompts to the space of target prompt. It is optimized by an Adam optimizer with the following hyperparameters: a learning rate of 1e-4, a weight decay of 0.1, a batch size of 13 (all source prompts are passed to the universal projector at once), with no warm up steps and no learning rate scheduler. This set of hyperparameters is grid-searched on CIFAR100 and extended as-is to all of the target tasks. Specifically, after training the universal projector with a LR-WD value pair, we evaluate the Kendall's coefficient between the ground-truth projection transfer rank and the rank of cosine-similarity between the target prompt and the projected source prompts. We train the universal projector on three different random seeds: [42, 44, 100] and report the grid-search results in Table 7.

Table 7: Grid-search results on CIFAR100 for the hyperparameters of the universal projector.

| ↓LR | WD→ | 0 | 0.1 | 0.01 | 0.001 | 0.0001 |
|---|---|---|---|---|---|
| 0.001 | 52.14±10.33 | 36.75±1.21 | 36.75±1.21 | 50.43±6.04 | 52.14±10.33 |
| 0.0005 | 80.34±8.46 | 45.30±1.21 | 73.50±8.46 | 80.34±8.46 | 80.34±8.46 |
| 0.0001 | 81.20±4.36 | **82.91±1.21** | 81.20±3.20 | 82.05±4.19 | 81.20±4.36 |
| 0.00005 | 76.92±0.00 | 75.21±1.21 | 76.92±0.00 | 76.92±0.00 | 76.92±0.00 |
| 0.00001 | 50.43±14.85 | 38.46±26.73 | 50.43±14.85 | 50.43±14.85 | 50.43±14.85 |

When the learning rate is set as 0.0001 and weight decay is set as 0.1, the universal projector gives the best cross-modality prompt transferability ranking score. Therefore, this set of hyperparameters is used on every target task. We need to emphasize that although performing grid-search on every target task would give better ranking scores, it is not right to do that on every task. As in practical, we need to know the transferability ranking scores before knowing the ground-truth transfer performance. Using the ground-truth transfer performance as the reference for conducting grid-search is paradoxical in practical scenarios: if we already have the ground-truth transfer performance, why would we even need to estimate the transferability.

# D DETAILED RESULTS

## D.1 CROSS-MODALITY PROMPT TRANSFER

### D.1.1 FROZEN PROMPT TRANSFER & PROJECTION TRANSFER

The detailed results are in Table 14. Note that all source-target pairs are ran on three random seeds: $[42, 44, 100]$, and their best results are reported in the format of mean±std.

### D.1.2 ATTENTION TRANSFER

For attention transfer, we release the training recipe and best accuracy in the format of mean±std on each target CV task in Table 8. Similarly, all experiments are ran on three random seeds: $[42, 44, 100]$.

Table 8: Hyper-parameters for attention transfer. The *Source* column shows the source prompt used for transfer while $l_s$ indicates the concentrated source prompt length. Note that on each target task, the Adam optimizer with a batch size of 64 is adopted. The target prompt length is always set as $(100 - l_s)$ on every target task.

| Task | Source | $l_s$ | LR | LR Warm-Up | LR Decay | Weight Decay | Accuracy |
|---|---|---|---|---|---|---|---|
| Caltech101 | SNLI | 80 | 0.005 | - | - | 0.001 | 90.09±0.28 |
| CIFAR100 | QQP | 60 | 0.005 | - | - | 0.001 | 81.72±0.34 |
| DTD | MNLI | 90 | 0.005 | - | - | 0.001 | 68.92±0.24 |
| Flowers102 | MNLI | 60 | 0.01 | 10 | Cosine | 0.001 | 98.37±0.12 |
| Pets | MNLI | 90 | 0.01 | 10 | Cosine | 0.001 | 89.42±0.08 |
| Sun397 | SNLI | 50 | 0.005 | - | - | 0.001 | 52.57±0.31 |
| SVHN | SNLI | 60 | 0.01 | 10 | Cosine | 0.001 | 82.76±0.24 |
| Patch Camelyon | SNLI | 80 | 0.1 | 10 | Cosine | 0.0001 | 82.08±1.42 |
| Resisc45 | MNLI | 100 | 0.01 | 10 | Cosine | 0.001 | 82.92±0.33 |
| EuroSat | SNLI | 100 | 0.01 | 10 | Cosine | 0.001 | 96.17±0.16 |
| Retinopathy | QNLI | 20 | 1 | 10 | Cosine | 0 | 75.30±0.44 |
| DMLab | QQP | 80 | 0.005 | 10 | Cosine | 0.01 | 44.85±0.31 |
| Kitti | QNLI | 70 | 0.005 | - | - | 0.001 | 77.37±0.40 |
| SmallNORB/azi | QQP | 100 | 0.001 | 10 | Cosine | 0.0001 | 24.34±1.82 |
| SmallNORB/ele | SNLI | 100 | 0.0005 | 10 | Cosine | 0 | 36.26±1.04 |
| dSprites/loc | SNLI | 80 | 0.005 | 10 | Cosine | 0.01 | 73.13±0.40 |
| dSprites/ori | SNLI | 90 | 0.005 | 10 | Cosine | 0.001 | 43.16±1.60 |
| Clevr/dist | MNLI | 90 | 0.001 | 10 | Cosine | 0 | 60.75±0.24 |
| Clevr/count | SNLI | 100 | 0.001 | 10 | Cosine | 0 | 67.92±0.51 |

## D.2 CROSS-MODALITY PROMPT TRANSFERABILITY ESTIMATION

### D.2.1 BASELINE METHOD: COSINE SIMILARITY BETWEEN AVERAGE PROMPTS

We show the best (SmallNORB/azi) and worst (Kitti) ranking results of the baseline method: Avg Cos in Table 9.

### D.2.2 BASELINE METHOD: ON

Similarly, the best (Flowers102) and worst (Resisc45) ranking results of ON are shown in Table 10.

### D.2.3 BASELINE METHOD: PRETRAINING DATA VOLUME

For this method, we simply list the volume of pretraining data of each source task in Table 11

### D.2.4 MODALITY AND TASK GAP

The best (Clevr/count) and worst (Retinopathy) ranking results are shown in Table 12.

Table 9: The cross-modality prompt transferability ranking results of Avg Cos on SmallNORB/azi (left) and Kitti (Right). *T Score* is the corresponding transferability score of each source prompt, predicted by the prompt transferability estimation method.

| Rank | SmallNORB/azi | | | | Kitti | | | |
|---|---|---|---|---|---|---|---|---|
| | Estimated | T Score | Ground Truth | Accuracy | Estimated | T Score | Ground Truth | Accuracy |
| 1 | sst2 | 0.042 | qqp | 23.61 | laptop | 0.033 | qnli | 76.98 |
| 2 | tweet | 0.040 | qnli | 23.36 | deontology | 0.027 | mnli | 76.84 |
| 3 | qnli | 0.039 | sst2 | 22.72 | movie | 0.010 | qqp | 76.56 |
| 4 | mrpc | 0.037 | deontology | 22.51 | imdb | 0.007 | snli | 76.47 |
| 5 | mnli | 0.036 | snli | 21.72 | restaurant | 0.001 | sst2 | 75.86 |
| 6 | qqp | 0.033 | tweet | 21.63 | sst2 | -0.001 | tweet | 75.25 |
| 7 | deontology | 0.029 | imdb | 21.50 | mrpc | -0.006 | deontology | 74.31 |
| 8 | justice | 0.006 | mnli | 21.05 | justice | -0.014 | justice | 72.53 |
| 9 | imdb | -0.002 | justice | 20.74 | snli | -0.032 | mrpc | 71.73 |
| 10 | snli | -0.003 | mrpc | 19.80 | mnli | -0.035 | imdb | 71.68 |
| 11 | laptop | -0.019 | restaurant | 18.62 | qqp | -0.039 | restaurant | 69.81 |
| 12 | restaurant | -0.046 | laptop | 16.82 | tweet | -0.046 | movie | 66.62 |
| 13 | movie | -0.049 | movie | 16.35 | qnli | -0.077 | laptop | 65.92 |

Table 10: The cross-modality prompt transferability ranking results of ON on Flowers102 (left) and Resisc45 (Right).

| Rank | Flowers102 | | | | Resisc45 | | | |
|---|---|---|---|---|---|---|---|---|
| | Estimated | T Score | Ground Truth | Accuracy | Estimated | T Score | Ground Truth | Accuracy |
| 1 | snli | 0.429 | mnli | 98.22 | deontology | 0.448 | snli | 82.01 |
| 2 | qqp | 0.429 | qqp | 98.20 | laptop | 0.392 | mnli | 81.92 |
| 3 | qnli | 0.407 | snli | 98.19 | movie | 0.382 | qqp | 81.54 |
| 4 | mnli | 0.390 | qnli | 98.04 | mrpc | 0.373 | qnli | 81.16 |
| 5 | sst2 | 0.376 | imdb | 97.99 | tweet | 0.366 | sst2 | 80.86 |
| 6 | imdb | 0.372 | tweet | 97.94 | sst2 | 0.359 | tweet | 80.07 |
| 7 | deontology | 0.364 | sst2 | 97.93 | justice | 0.340 | imdb | 79.63 |
| 8 | tweet | 0.352 | justice | 97.82 | qnli | 0.312 | justice | 79.04 |
| 9 | justice | 0.329 | deontology | 97.69 | imdb | 0.312 | deontology | 78.08 |
| 10 | movie | 0.328 | restaurant | 97.69 | restaurant | 0.307 | mrpc | 77.00 |
| 11 | laptop | 0.311 | mrpc | 97.59 | qqp | 0.303 | movie | 76.95 |
| 12 | mrpc | 0.299 | laptop | 97.48 | mnli | 0.283 | restaurant | 76.78 |
| 13 | restaurant | 0.292 | movie | 97.45 | snli | 0.278 | laptop | 76.16 |

Table 11: Pretraining data volume of each source NLP task.

| Source Task | imdb | sst2 | laptop | restaurant | movie | tweet | mnli | qnli | snli | deontology | justice | qqp | mrpc |
|---|---|---|---|---|---|---|---|---|---|---|---|---|---|
| Data Volume | 25000 | 67349 | 3045 | 3041 | 1600 | 45389 | 392702 | 104743 | 549367 | 18164 | 21791 | 363846 | 3668 |

Table 12: The cross-modality prompt transferability ranking results of our method ($\mathcal{G}_M$ & $\mathcal{G}_T$) on Clevr/count (left) and Retinopathy (Right).

| Rank | Clevr/count | | | | Retinopathy | | | |
|---|---|---|---|---|---|---|---|---|
| | Estimated | T Score | Ground Truth | Accuracy | Estimated | T Score | Ground Truth | Accuracy |
| 1 | snli | -0.619 | snli | 62.60 | qqp | -0.213 | qnli | 74.76 |
| 2 | mnli | -0.675 | mnli | 58.52 | snli | -0.269 | qqp | 74.76 |
| 3 | qqp | -0.698 | qnli | 57.67 | qnli | -0.311 | snli | 74.55 |
| 4 | qnli | -0.744 | qqp | 56.14 | mnli | -0.313 | sst2 | 74.53 |
| 5 | sst2 | -0.776 | sst2 | 55.24 | sst2 | -0.659 | mnli | 74.49 |
| 6 | tweet | -0.980 | tweet | 55.01 | tweet | -0.820 | tweet | 74.42 |
| 7 | deontology | -1.021 | deontology | 53.02 | deontology | -0.935 | laptop | 74.40 |
| 8 | justice | -1.291 | imdb | 52.77 | justice | -1.122 | deontology | 74.37 |
| 9 | imdb | -1.355 | justice | 52.27 | imdb | -1.144 | movie | 74.37 |
| 10 | mrpc | -1.463 | mrpc | 51.75 | mrpc | -1.258 | imdb | 74.36 |
| 11 | restaurant | -1.844 | restaurant | 48.72 | restaurant | -1.425 | mrpc | 74.29 |
| 12 | laptop | -1.916 | laptop | 45.34 | laptop | -1.460 | justice | 74.24 |
| 13 | movie | -1.962 | movie | 44.38 | movie | -1.476 | restaurant | 74.24 |

## D.2.5 GROUND-TRUTH GAP VALUES

The ground-truth values of the combined modality and task gaps are in Table 13.

Table 13: The ground-truth values of the combined modality and task gap.

|  | IMDB | SST2 | Laptop | Restaurant | Movie | Tweet | MNLI | QNLI | SNLI | Deontology | Justice | QQP | MRPC |
|---|---|---|---|---|---|---|---|---|---|---|---|---|---|
| Caltech101 | 1.116 | 0.759 | 1.284 | 1.269 | 1.320 | 0.900 | -0.141 | 0.239 | -0.180 | 1.053 | 1.110 | 0.082 | 1.121 |
| CIFAR100 | 1.455 | 0.721 | 1.925 | 1.878 | 1.958 | 0.988 | 0.120 | 0.118 | 0.077 | 1.088 | 1.391 | 0.074 | 1.553 |
| DTD | 1.344 | 1.061 | 1.472 | 1.464 | 1.485 | 1.179 | 0.101 | 0.538 | 0.063 | 1.249 | 1.311 | 0.393 | 1.375 |
| Flowers102 | 1.412 | 1.094 | 1.561 | 1.551 | 1.579 | 1.224 | 0.128 | 0.561 | 0.127 | 1.300 | 1.390 | 0.374 | 1.450 |
| Pets | 1.302 | 0.868 | 1.545 | 1.519 | 1.572 | 1.011 | -0.008 | 0.264 | -0.025 | 1.146 | 1.316 | 0.077 | 1.375 |
| Sun397 | 1.286 | 0.881 | 1.454 | 1.440 | 1.488 | 1.050 | -0.073 | 0.347 | -0.109 | 1.176 | 1.283 | 0.124 | 1.329 |
| SVHN | 1.640 | 0.801 | 2.599 | 2.462 | 2.650 | 1.032 | 0.663 | 0.591 | 0.657 | 1.214 | 1.546 | 0.671 | 1.859 |
| Patch Camelyon | 1.962 | 1.685 | 2.058 | 2.064 | 2.080 | 1.788 | 0.513 | 1.164 | 0.497 | 1.857 | 1.928 | 0.904 | 1.995 |
| Resisc45 | 1.310 | 0.836 | 1.553 | 1.530 | 1.583 | 0.974 | 0.003 | 0.223 | -0.032 | 1.125 | 1.287 | 0.074 | 1.364 |
| EuroSAT | 1.753 | 0.873 | 2.525 | 2.415 | 2.565 | 1.160 | 0.396 | 0.447 | 0.393 | 1.309 | 1.660 | 0.464 | 1.972 |
| Retinopathy | 1.144 | 0.659 | 1.460 | 1.425 | 1.476 | 0.820 | 0.313 | 0.311 | 0.269 | 0.935 | 1.122 | 0.213 | 1.258 |
| DMLab | 0.924 | 0.520 | 1.227 | 1.152 | 1.387 | 0.659 | -0.474 | -0.045 | -0.463 | 0.752 | 0.899 | -0.271 | 0.975 |
| KITTI | 0.479 | 0.214 | 0.713 | 0.639 | 0.874 | 0.304 | -0.608 | -0.187 | -0.602 | 0.377 | 0.470 | -0.390 | 0.512 |
| SmallNORB/azi | 2.784 | 1.870 | 3.546 | 3.466 | 3.607 | 2.132 | 0.721 | 1.185 | 0.715 | 2.307 | 2.697 | 1.084 | 2.951 |
| SmallNORB/ele | 2.695 | 1.702 | 3.514 | 3.414 | 3.587 | 1.981 | 0.527 | 1.002 | 0.523 | 2.185 | 2.583 | 0.896 | 2.882 |
| dSprites/loc | 3.445 | 2.750 | 3.907 | 3.853 | 3.930 | 3.011 | 1.128 | 2.024 | 1.127 | 3.125 | 3.428 | 1.766 | 3.586 |
| dSprites/ori | 3.439 | 2.725 | 3.895 | 3.838 | 3.922 | 2.997 | 1.107 | 1.998 | 1.152 | 3.144 | 3.420 | 1.707 | 3.568 |
| Clevr/dist | 1.309 | 0.737 | 1.915 | 1.834 | 1.944 | 0.907 | 0.579 | 0.665 | 0.591 | 1.059 | 1.287 | 0.661 | 1.475 |
| Clevr/count | 1.355 | 0.776 | 1.916 | 1.844 | 1.962 | 0.980 | 0.675 | 0.744 | 0.619 | 1.021 | 1.291 | 0.698 | 1.463 |

Table 14: The detailed results of Frozen Prompt Transfer (gray) and Projection Transfer (Blue). The accuracy marked in red indicates the best source prompts for each target task.

| | Caltech101 | CIFAR100 | DTD | Flowers102 | Pets | Sun397 | SVHN | Mean | Patch Camelyon | Resisc45 | EuroSAT | Retinopathy | Mean | DMLab | KITTI | SmallNORB/azi | SmallNORB/ele | dSprites/loc | dSprites/ori | Clevr/dist | Clevr/count | Mean |
|---|---|---|---|---|---|---|---|---|---|---|---|---|---|---|---|---|---|---|---|---|---|---|
| Linear | 86.25±0.03 | 65.26±0.06 | 64.40±0.13 | 97.27±0.01 | 86.55±0.03 | 51.22±0.01 | 36.25±0.09 | 69.60 | 79.02±0.36 | 69.54±0.08 | 89.34±0.32 | 74.31±0.13 | 78.05 | 33.83±0.20 | 60.10±0.45 | 11.92±0.07 | 20.90±0.35 | 12.67±0.52 | 20.99±0.01 | 31.21±0.16 | 36.29±0.04 | 28.49 |
| Xavier | 82.31±0.27 | 61.60±0.89 | 59.41±0.44 | 95.51±0.27 | 84.16±0.39 | 45.98±0.41 | 37.62±0.18 | 66.66 | 80.34±0.20 | 66.62±0.64 | 90.13±0.76 | 74.12±0.08 | 77.80 | 33.09±0.85 | 59.54±0.65 | 11.84±0.12 | 18.40±0.72 | 10.50±0.37 | 15.98±0.19 | 33.11±0.31 | 37.45±0.07 | 27.49 |
| IMDB | 82.37±0.13 | 62.38±0.02 | 59.82±0.05 | 96.01±0.05 | 84.22±0.10 | 46.94±0.13 | 37.31±0.09 | 67.01 | 79.73±0.11 | 67.15±0.09 | 90.52±0.54 | 74.02±0.01 | 77.86 | 33.16±0.22 | 58.09±0.20 | 11.65±0.33 | 19.86±0.56 | 11.66±0.14 | 16.41±0.01 | 31.66±0.57 | 37.12±0.08 | 27.45 |
| SST2 | 83.63±0.05 | 61.85±0.04 | 60.64±0.12 | 96.09±0.03 | 84.73±0.09 | 47.94±0.14 | 37.18±0.03 | 67.44 | 79.22±0.08 | 69.05±0.07 | 90.10±0.36 | 73.99±0.16 | 78.09 | 33.54±0.10 | 59.92±0.58 | 11.35±0.26 | 20.01±0.41 | 13.13±0.12 | 15.73±0.02 | 32.00±0.26 | 35.93±0.07 | 27.70 |
| Laptop | 82.31±0.06 | 62.38±0.02 | 59.20±0.04 | 95.72±0.01 | 84.20±0.07 | 46.24±0.14 | 38.12±0.06 | 66.88 | 80.22±0.14 | 65.87±0.00 | 89.62±0.78 | 73.92±0.14 | 77.41 | 33.14±0.13 | 59.26±0.29 | 11.70±0.11 | 19.72±0.65 | 10.78±0.13 | 15.67±0.05 | 32.28±0.75 | 37.25±0.07 | 27.48 |
| Restaurant | 82.44±0.13 | 62.66±0.03 | 59.98±0.18 | 95.97±0.03 | 84.73±0.04 | 46.32±0.11 | 37.80±0.07 | 67.13 | 80.33±0.11 | 66.91±0.15 | 90.82±0.51 | 74.04±0.02 | 78.03 | 33.96±0.43 | 59.68±0.58 | 11.68±0.20 | 20.16±0.67 | 9.57±0.19 | 15.77±0.03 | 32.08±0.19 | 37.42±0.05 | 27.54 |
| Movie | 82.28±0.05 | 62.24±0.06 | 59.63±0.24 | 95.80±0.02 | 84.22±0.00 | 46.41±0.07 | 37.94±0.04 | 66.93 | 80.28±0.12 | 66.73±0.08 | 90.40±0.63 | 73.75±0.09 | 77.79 | 33.41±0.15 | **61.13**±0.37 | 11.59±0.16 | 19.98±1.16 | 10.41±0.13 | 16.17±0.01 | 31.36±0.60 | **37.73**±0.03 | 27.72 |
| Tweet | 83.06±0.19 | 62.83±0.07 | 60.20±0.05 | 96.29±0.02 | 84.70±0.03 | 48.20±0.14 | 37.32±0.11 | 67.51 | 79.06±0.20 | 68.84±0.04 | 90.38±0.41 | 74.10±0.07 | 78.10 | 33.80±0.12 | 60.57±0.63 | 11.52±0.12 | **20.15**±0.51 | 14.14±0.17 | **17.17**±0.03 | **32.73**±0.20 | 36.33±0.04 | **28.30** |
| MNLI | 84.56±0.32 | 64.92±0.17 | **63.14**±0.24 | 96.29±0.02 | 86.20±0.15 | 48.94±0.07 | 37.65±0.45 | 68.81 | **80.36**±0.13 | **72.08**±0.02 | 90.79±0.44 | **74.14**±0.06 | **79.34** | 34.80±0.30 | 58.09±0.41 | 9.98±0.29 | 18.57±0.41 | 12.91±0.16 | 13.64±0.02 | 32.66±0.22 | 35.96±0.03 | 27.08 |
| QNLI | 84.27±0.09 | 63.93±0.12 | 60.62±0.05 | 96.33±0.03 | 85.75±0.11 | 48.75±0.02 | 37.66±0.08 | 68.19 | 79.93±0.13 | 71.04±0.09 | **90.83**±0.73 | 74.01±0.20 | 78.95 | 33.64±0.14 | 60.52±0.86 | 11.02±0.11 | 19.97±0.76 | 11.80±0.25 | 14.76±0.02 | 31.66±0.39 | 35.92±0.04 | 27.41 |
| SNLI | **84.75**±0.08 | **65.56**±0.02 | 62.59±0.10 | **96.55**±0.01 | **86.20**±0.10 | **49.46**±0.05 | 37.16±0.40 | **68.90** | 80.32±0.08 | 71.32±0.09 | 90.00±0.66 | 73.94±0.20 | 78.90 | **34.82**±0.13 | 58.60±0.80 | 10.85±0.42 | 18.64±0.20 | 11.58±0.28 | 14.30±0.00 | 31.53±0.34 | 36.68±0.04 | 27.13 |
| Deontology | 82.97±0.16 | 62.21±0.06 | 59.33±0.02 | 95.84±0.05 | 84.48±0.05 | 48.10±0.11 | 37.83±0.10 | 67.25 | 78.90±0.15 | 68.30±0.04 | 90.24±0.15 | 74.10±0.02 | 77.89 | 33.32±0.23 | 60.80±1.33 | 11.43±0.16 | 19.68±0.41 | 13.90±0.33 | 16.89±0.01 | 32.18±0.56 | 36.46±0.01 | 28.08 |
| Justice | 82.47±0.08 | 62.52±0.04 | 59.04±0.09 | 95.54±0.05 | 84.53±0.05 | 47.17±0.13 | 37.21±0.05 | 66.93 | 79.35±0.07 | 67.23±0.10 | 90.06±0.53 | 73.99±0.08 | 77.66 | 33.06±0.38 | 59.54±0.75 | 11.78±0.20 | 19.57±0.45 | **15.41**±0.21 | 16.69±0.02 | 31.86±0.29 | 37.02±0.01 | 28.12 |
| QQP | 84.28±0.12 | 62.49±0.06 | 62.39±0.19 | 96.45±0.01 | 86.08±0.16 | 49.08±0.09 | **38.39**±0.03 | 68.80 | 79.66±0.15 | 71.41±0.08 | 90.62±0.34 | 74.05±0.13 | 78.94 | 34.21±0.13 | 58.13±1.15 | 11.19±0.04 | 19.87±0.69 | 10.86±0.10 | 15.03±0.00 | 32.07±0.53 | 35.76±0.11 | 27.14 |
| MRPC | 82.50±0.13 | 62.49±0.03 | 58.48±0.15 | 95.09±0.05 | 84.14±0.19 | 46.34±0.11 | 38.04±0.08 | 66.73 | 78.86±0.11 | 66.27±0.10 | 88.90±0.05 | 73.96±0.17 | 77.00 | 33.46±0.15 | 61.04±0.30 | **12.07**±0.70 | 19.33±0.92 | 11.81±0.04 | 16.72±0.00 | 31.97±0.81 | 37.30±0.05 | 27.96 |
| VPT | 88.73±0.17 | 78.03±0.17 | 65.73±0.11 | 97.75±0.11 | 87.93±0.25 | 50.21±0.09 | 71.26±1.15 | 77.09 | 80.84±0.58 | 75.70±0.08 | 92.90±0.42 | 73.95±0.07 | 80.85 | 39.81±1.20 | 72.15±2.81 | 17.36±0.06 | 22.83±0.37 | 72.94±1.37 | 30.94±2.25 | 57.59±0.29 | 49.96±0.58 | 45.45 |
| Xavier | 87.66±0.05 | 74.28±0.57 | 65.12±0.52 | 97.58±0.06 | 87.67±0.19 | 51.02±0.19 | 66.47±1.52 | 75.69 | 79.62±0.44 | 76.66±0.15 | 93.33±0.13 | 74.24±0.05 | 80.96 | 37.59±0.39 | 66.95±1.80 | 16.47±0.17 | 23.21±0.35 | 62.25±1.43 | 30.25±1.60 | 53.04±0.15 | 45.62±0.72 | 41.92 |
| IMDB | 88.31±0.20 | 78.18±0.24 | 64.97±0.11 | 97.99±0.04 | 88.42±0.13 | 51.70±0.20 | 75.04±1.58 | 77.80 | 80.09±0.92 | 79.63±0.94 | 94.11±0.09 | 74.36±0.05 | 82.05 | 40.04±0.25 | 71.68±1.40 | 21.50±0.25 | 24.87±0.51 | 68.55±0.58 | 38.42±1.70 | 52.77±0.57 | 46.78 |
| SST2 | 88.40±0.05 | 79.52±0.58 | 66.60±0.08 | 97.93±0.09 | 88.57±0.17 | 52.14±0.07 | 78.35±0.55 | 78.79 | 79.73±0.26 | 80.86±0.50 | 95.22±0.23 | 74.53±0.10 | 82.59 | 41.95±0.68 | 75.86±0.26 | 22.72±0.86 | 26.12±0.78 | 66.98±0.75 | 39.74±0.43 | 57.20±0.21 | 55.24±0.58 | 48.23 |
| Laptop | 87.78±0.14 | 75.61±0.73 | 64.54±0.13 | 97.48±0.04 | 87.64±0.14 | 50.91±0.16 | 63.63±0.99 | 75.37 | 79.39±0.46 | 76.16±0.60 | 93.32±0.43 | 74.40±0.11 | 80.82 | 37.22±0.42 | 65.92±0.85 | 16.82±0.52 | 23.98±0.47 | 57.97±5.36 | 30.62±2.36 | 52.46±3.55 | 45.34±1.59 | 41.29 |
| Restaurant | 88.02±0.17 | 76.20±0.72 | 64.91±0.17 | 97.69±0.14 | 87.86±0.08 | 51.10±0.10 | 68.12±0.09 | 76.27 | 79.44±0.93 | 76.78±0.17 | 93.46±0.20 | 74.24±0.05 | 80.98 | 37.86±0.21 | 69.81±1.79 | 18.62±1.02 | 23.86±0.45 | 60.73±5.62 | 36.51±0.84 | 53.58±0.91 | 48.72±0.73 | 43.71 |
| Movie | 87.54±0.32 | 75.49±0.58 | 64.58±0.08 | 97.45±0.13 | 87.47±0.18 | 51.12±0.19 | 64.30±1.04 | 75.42 | 78.95±0.43 | 76.95±0.39 | 93.22±0.37 | 74.37±0.03 | 80.87 | 36.88±0.31 | 66.62±0.24 | 16.35±0.20 | 23.40±0.61 | 62.74±3.49 | 32.72±2.06 | 53.93±0.64 | 44.38±0.21 | 42.13 |
| Tweet | 88.28±0.05 | 78.94±0.44 | 66.24±0.17 | 97.94±0.05 | 88.64±0.23 | 52.15±0.18 | 78.35±0.77 | 78.65 | 80.17±0.71 | 80.07±0.45 | 95.17±0.29 | 74.42±0.11 | 82.46 | 41.53±0.54 | 75.25±0.58 | 21.63±0.49 | 26.26±0.82 | 63.91±2.16 | 41.32±2.81 | 56.55±1.14 | 55.01±0.99 | 47.68 |
| MNLI | 88.67±0.09 | 80.49±0.12 | **68.44**±0.14 | **98.22**±0.02 | **89.02**±0.02 | **52.64**±0.03 | 82.04±0.65 | 79.93 | 80.91±0.44 | 81.92±0.55 | 95.66±0.10 | 74.49±0.20 | **83.25** | 43.37±0.92 | 76.84±0.35 | 21.05±0.65 | 26.49±0.23 | 70.33±1.77 | 41.72±1.34 | **58.44**±0.58 | 58.52±0.55 | 49.60 |
| QNLI | 88.37±0.04 | 80.12±0.38 | 67.27±0.29 | 98.04±0.01 | 88.46±0.23 | 52.18±0.34 | 81.01±1.07 | 79.35 | 79.75±0.63 | 81.16±0.71 | **95.77**±0.08 | 74.76±0.12 | 82.86 | 43.45±0.75 | **76.98**±0.58 | 23.36±0.65 | 26.33±0.82 | 65.70±0.28 | 41.55±0.27 | 57.21±0.56 | 57.67±0.77 | 49.03 |
| SNLI | **89.04**±0.05 | 80.77±0.19 | 68.33±0.52 | 98.19±0.07 | 88.62±0.19 | 52.27±0.15 | **82.58**±0.73 | **79.97** | 80.60±0.69 | **82.01**±0.12 | 95.75±0.04 | 74.55±0.30 | 83.23 | 43.11±0.57 | 76.47±1.03 | 21.72±0.73 | 26.81±0.86 | **72.46**±2.48 | **42.35**±0.47 | 57.91±0.98 | **62.60**±3.09 | **50.43** |
| Deontology | 88.40±0.06 | 78.26±0.36 | 65.32±0.23 | 97.69±0.07 | 87.85±0.27 | 51.81±0.11 | 76.50±0.55 | 77.98 | 79.61±0.66 | 78.08±0.54 | 94.01±0.22 | 74.37±0.07 | 81.52 | 41.43±0.53 | 74.31±0.66 | 22.51±0.53 | 26.04±0.39 | 63.31±3.86 | 39.11±0.39 | 56.36±1.35 | 53.02±2.01 | 47.01 |
| Justice | 88.42±0.50 | 77.87±0.11 | 64.86±0.07 | 97.82±0.12 | 87.82±0.27 | 51.08±0.36 | 76.56±0.77 | 77.78 | 79.59±0.32 | 79.04±1.42 | 94.28±0.34 | 74.44±0.16 | 81.79 | 40.69±1.25 | 72.53±2.26 | 20.74±0.25 | 25.44±1.33 | 67.06±2.04 | 38.11±1.17 | 56.00±1.07 | 52.27±0.84 | 46.61 |
| QQP | 88.66±0.20 | **80.98**±0.18 | 67.91±0.22 | 98.20±0.03 | 88.41±0.30 | 52.27±0.07 | 82.45±1.10 | 79.84 | **80.98**±0.18 | 81.54±0.04 | 95.54±0.27 | **74.76**±0.19 | 83.21 | **43.61**±0.24 | 76.56±0.40 | **23.61**±1.37 | **28.27**±0.69 | 70.73±1.80 | 40.85±2.08 | 57.57±0.96 | 56.14±2.47 | 49.67 |
| MRPC | 88.26±0.27 | 76.92±0.31 | 64.63±0.23 | 97.59±0.06 | 87.67±0.07 | 50.93±0.36 | 73.16±0.90 | 77.02 | 79.12±1.21 | 77.00±0.24 | 93.53±0.23 | 74.29±0.06 | 80.99 | 38.59±1.16 | 71.73±0.94 | 19.80±0.43 | 24.72±0.53 | 66.23±0.98 | 37.99±1.24 | 55.69±0.38 | 51.75±2.23 | 45.81 |

# E  DISCUSSIONS

**Discussion on our choice of ViT pretrained checkpoint.**    We discuss our choice on the supervised ImageNet-21k ViT checkpoint (ViT-IN21k for short) from two perspectives:

- **Why not a checkpoint pretrained on a different dataset or by a different pretraining objective**: we need to demonstrate how significantly can cross-modality prompt transfer boost prompt tuning. In the meantime, visual prompt tuning was initially experimented on the ViT-IN21k. Therefore, to make the comparison meaningful, we select the ViT-IN21k checkpoint.

- **Why not a language-aligned ViT checkpoint**: we need to first demonstrate that a model, even pretrained without any textual knowledge (e.g. the ViT-IN21k checkpoint), can benefit from transferring prompts pretrained on pure textual data. Second, the source prompts are trained with language models on language tasks. The language models are not aligned with any variations of ViT checkpoint and the language tasks are not aligned with any image tasks. We believe that a language-aligned ViT checkpoint should be applied to a situation where there are connections across modalities. However in our cases, the two modalities are isolated. Therefore, we choose the ViT-IN21k checkpoint instead of a language-aligned ViT checkpoint.

**Real-world applications of cross-modality prompt transfer.**    In terms of real-world scenarios, cross-modality prompt transfer can be of help from two aspects: (i) improve the prompt tuning performance on data-scarce tasks or privacy sensitive tasks (for example, medical diagnosis or business analysis), as it transfers abundant source knowledge without depending on the source data. (ii) improve the prompt tuning performance on tasks that would benefit from combining knowledge across text and vision domains, especially when one modality has abundant labelled data or provides complementary insights. For example, it can enhance medical image analysis by transferring prompts pretrained on text-based medical knowledge, support vision systems used in autonomous vehicles by transferring prompts trained on traffic rule texts, etc.

**Why would the source prompts pretrained on NLP tasks transferable to other modality tasks?**
We analysis this problem through two perspectives: the source prompts themselves, and the language model used to produce the source prompts. (i) Transferable source prompts are typically pretrained on a huge volume of NLP data. As a result, these prompts should contain valuable semantics that are high-level enough to be extendable to other modalities. Moreover, they should also have a generalizability powerful enough to generalize to an unseen modality. (ii) On the other hand, language models trained on large and diverse datasets, learn representations that capture concepts, relationships, and structures beyond text alone. These abstract features may generalize well to other modalities, where the prompts can act as conceptual "anchors" that "carry" the representations across modalities.

**Which target tasks are more likely to benefit from cross-modality prompt transfer?**    In our main paper, Section 4.2, we draw a conclusion that "Prompt tuning performance on target tasks that already benefit a lot from prompt tuning is likely to be further boosted by cross-modality prompt transfer". Here we discuss the intuition behind this conclusion: If a target task benefits a lot from prompt tuning (i.e. has a higher prompt tuning performance than linear probing), this indicates that this target task requires the pretrained model to learn new features, because simple linear probing on its original features does not help much. In another word, such a target task would require the pretrained model to learn extra knowledge. Therefore, why cross-modality prompt transfer excels on such a target task is easy to understand: it introduces more knowledge to the target task than vanilla prompt tuning. As vanilla prompt tuning only learns target knowledge while cross-modality prompt transfer learns both the source and target knowledge. In conclusion, if a target task requires the pretrained model to grasp a lot of new features, then it would be a more transferable target task, and vice versa.

**The improvements made by projection transfer is proven to be statistically significant.**    We prove this statistical significance following the below steps: (i) Define our null hypothesis: the improvements are not introduced by transferring the source prompts. (ii) We choose one-sided test as

the effects will be in one direction (i.e. contribute to the increase of the improvements) (iii) Collect data: Group 1 (Conducting projection transfer on all 19 target tasks with a randomly initialized prompt): In this case, we have only one "source" prompt and 19 target tasks, which makes the total number of observations to be 19. Among these 19 observations, only 5 show improvements over vanilla prompt tuning, resulting in a percentage of 26.32%. (The data comes from the first row of Figure 3b.2) Group 2 (Conducting projection transfer on all 19 target tasks with all 13 source tasks): In this case, we have 13 source prompts and 19 target tasks, resulting in a total number of 19*13=247 observations. Among these 247 observations, 158 show improvements over vanilla prompt tuning, with a percentage of 63.97%. (The data comes from the remaining rows of Figure 3b.2) (iv) Calculate p-value: The final p-value is calculated to be 0.02%. That is, given the results we observed, there's only a chance of 0.02% that the improvements are not introduced by cross-modality prompt transfer. In another word, we are 99.98% certain that the improvements are statistical significant: they are caused by cross-modality prompt transfer, instead of error or random chances.

**Limitations and future works.** Our work contributes to the field by the verification of the feasibility of cross-modality prompt transfer, the conceptualization of the transferability estimation metric and a more effective cross-modality prompt transfer method. Besides its contributions, it also comes with the following limitations in the current stage: (i) The scope: only one model from each modality was involved in our experiments. Moreover, different prompt tuning variants (i.e. prefix-tuning) were not covered in our explorations. (ii) The dependence on trainable parameter of the proposed transferability estimation metric: this dependence could slightly increase the computation complexity of the metric. Given the current development of prompt transfer and the fact that we are the first work on the topic of cross-modality prompt transfer, these limitations are hard to avoid in the current stage. However, we are confident that these limitations will not affect the conclusions drawn in this paper.

In the future, besides breaking the limitations discussed above (i.e. extending the scope and reducing the complexity of the transferability estimation metric), it would be interesting to explore multi-source prompt fusion methods for cross-modality prompt transfer: how to fuse multiple text-pretrained prompts to make better utilization of the linguistic knowledge stored in the prompts. Our future work aims to further solidify our findings about cross-modality prompt transfer and further release the powers of the source prompts on data-scarce modalities.

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
