# OpenReview forum: "Release the Powers of Prompt Tuning: Cross-Modality Prompt Transfer"
_ICLR.cc/2025/Conference — ICLR 2025 Poster_

### Official Review · Reviewer_Ve1v · 2024-10-30

**Soundness:** 2
**Presentation:** 3
**Contribution:** 3
**Rating:** 8
**Confidence:** 4

**Summary:**

This work proposes to improve the performance of prompt tuning by exploring cross-modality prompt transfer, that is leveraging prompts pretrained on text modality to improve performance on few-shot image recognition. They first validate the effectiveness of cross-modality prompt transfer by empirical comparison between VPT, linear probing, frozen prompt transfer and projection transfer. They also propose an effective metric to estimate prompt transferability by combining modality gap and task gap scores. Finally, they propose the attention transfer which introduces the knowledge from source domain by an attention block. Experiments on VTAB-1K validate that attention transfer can improve existing SOTA method.

**Strengths:**

1. Most parts of the paper are clearly presented.
2. Promising results with extensive experiments.
3. Novel idea of cross-modality transfer instead of the intra-modality transfer.

**Weaknesses:**

My concern focuses on the effectiveness of the proposed method. (1) The proposed projection transfer and attention transfer have introduced extra parameters, the effect from extra parameters and source domain are not decomposed in the experiments in Fig.3. This makes it hard to judge if the source domain knowledge is really useful. The authors can further strengthen the experiments by 1) using random source prompt in projection transfer and attention transfer, which will make the number of learnable parameters match to that in experiments using source prompts from NLP domain, 2) using a source domain with many training data and gradually increasing the number of source training samples and inspecting if the source prompts learned with different number of source samples lead to increasing trend of improvement in projection transfer and attention transfer. (2) The effectiveness should be demonstrated using different language models, so as to validate the generalization ability. (3) The effectiveness should be demonstrated using different visual prompt tuning methods, including shallow and deep methods. (4) The accuracy on Natural subset decreases, making it hard to determine the effectiveness. Experiments on more datasets should be conducted to support the claim of this paper.

Besides, the meaning of designing a good metric estimating prompt transferability should be also clarified. GM & GT outperforms Vdata by 3 points with respect to the mean Kendall's coefficient. To what degree does such an improvement contribute to enhancing the transfer performance on VTAB-1k? The work "Qifan Wang, Yuning Mao, Jingang Wang, Hanchao Yu, Shaoliang Nie, Sinong Wang, Fuli Feng, Lifu Huang, Xiaojun Quan, Zenglin Xu, Dongfang Liu: APrompt: Attention Prompt Tuning for Efficient Adaptation of Pre-trained Language Models. EMNLP 2023: 9147-9160" also introduced attention into prompt tuning, the relationship of the proposed method to it should be also clarified.

**Questions:**

See the weaknesses.

---

> ### Author Response · Authors · 2024-11-21
> **Response to weakness 1**
>
> # **Weakness 1(if the source domain knowledge is really useful?)**
>
> Thank you deeply for your suggestions! We need to bring to your attention that the results of projection transfer using a random prompt are reported in **Figure 3b**.
>
> In **Figure 3**, the first row of all the heatmaps is obtained by conducting frozen prompt transfer or projection transfer using a random prompt. We name it “xavier” as the random prompt is randomly initialized using Xavier initialization [1] as was used in the original Visual Prompt Tuning paper [2].
>
> The relative performance reported in the left half of **Figure 3** (the a.1 and b.1 heatmaps) all sets the random prompt as the baseline comparison.
>
> In a.1, 61.5% of the transfer pairs (150 out of 247) can achieve better performance compared to conducting frozen prompt transfer using a random prompt.
>
> In b.1, 86.2% of the transfer pairs (213 out of 247) can achieve better performance compared to conducting projection transfer using a random prompt.
>
> We can see that the source knowledge is indeed helpful to the projection transfer performance, compared to a random prompt. As for attention transfer, it also adapts the source prompt by linear projection (the source prompt vectors are first linearly projected and then weighted-summed), like projection transfer. Given this similarity, we can make a reasonable conjecture that attention transfer with a random prompt will not strike a performance. Therefore, we think that the comparison in **Figure 3a.1 and b.1** is enough to demonstrate the importance of source knowledge stored in the pretrained source prompts.
>
> As for your suggestion 2): using a source domain with many training data and gradually increasing the number of source training samples and inspecting if the source prompts learned with different number of source samples lead to increasing trend of improvement in projection transfer and attention transfer.
>
> In our current experiments, we wish to keep our experimental setup to a source-data-free status. That is, all information we have about the source task is the weights trained on this task. We keep this setup as it's more realistic: in real-world scenarios, it is highly possible that only the weights instead of the data are shared across tasks, as data may contain privacy-sensitive information.
>
> Nonetheless, we believe your suggestion is a really meaningful exploration: apart from verifying the importance of source knowledge, we can draw additional conclusions about which types of NLP tasks are more favourable to the CV tasks by putting the number of samples in various types of NLP tasks at the same baseline. However, given our current experiment workload and the fact that we have verified the importance of source knowledge stored in the pretrained source prompts, we would leave such exploration to our future work. Thank you!
>
> [1] Glorot, Xavier, and Yoshua Bengio. "Understanding the difficulty of training deep feedforward neural networks." In Proceedings of the thirteenth international conference on artificial intelligence and statistics, pp. 249-256. JMLR Workshop and Conference Proceedings, 2010.
>
> [2] Jia, Menglin, Luming Tang, Bor-Chun Chen, Claire Cardie, Serge Belongie, Bharath Hariharan, and Ser-Nam Lim. "Visual prompt tuning." In European Conference on Computer Vision, pp. 709-727. Cham: Springer Nature Switzerland, 2022.

---

> ### Author Response · Authors · 2024-11-21
> **Response to weakness 2 and 3**
>
> # **Weakness 2 (The effectiveness should be demonstrated using different language models, so as to validate the generalization ability)**
>
> Our proposed projection transfer and attention transfer can be safely extended to prompts trained by different language models, even to an encoder-decoder language model like T5.
>
> Specifically, we conduct projection transfer and attention transfer on three target tasks: CIFAR100, EuroSat, and Kitti (we choose 1 each from the VTAB-Natural, VTAB-Specialized, and VTAB-Structured groups). Below are the results (the results are averaged over three runs):
>
> | | CIFAR100 | EuroSat | Kitti |
> |-|-|-|-|
> | Finetuning | 68.9 | 95.7 | 65.5 |
> | Visual Prompt Tuning | 78.0 | 92.9 | 72.2 |
> | SPT (original paper) | 78.9 | 94.9 | 68.8 |
> | Projection Transfer (RoBERTa) | 81.0 | 95.8 | 77.0 |
> | Attention Transfer (RoBERTa) | 81.7 | **96.2** | 77.4 |
> | Projection Transfer (BERT) | 81.6 | 95.8 | 77.7 |
> | Attention Transfer (BERT) | **82.1** | 95.9 | **78.0** |
> | Projection Transfer (T5) | 81.6 | 96.0 | 77.3 |
> | Attention Transfer (T5) | 81.9 | 96.0 | 77.9 |
>
> We can see that our projection transfer and attention transfer generalize well to source prompts trained by different language models. But how different languages impact the transfer performance and how to choose the best source prompt in a multi-language-model situation still needs more extensive experiments and in-depth analysis. We need to leave those to our future work.
>
>
>
> # **Weakness 3 (The effectiveness should be demonstrated using different visual prompt tuning methods, including shallow and deep methods)**
>
> Thank you for your suggestion.
>
> Since all the results reported in our paper were obtained through the shallow method, we hereby extend our attention transfer to the deep method. Specifically, for the input of each attention block in ViT, we prepend 10 prompts. Each 10 prompts are cooked by a prompt concentrator proposed in our attention transfer, concentrating a length 100 source prompt to length 10.
>
> For ViT-base, we use 12 prompt concentrators in total, one for each layer. But the source prompt is shared across all layers. We let the 12 prompt concentrators to decide how the source prompt should look like in each layer.
>
> Below are the results (the results are averaged over three runs):
>
> | | CIFAR100 | EuroSat | Kitti |
> |-|-|-|-|
> | Finetuning | 68.9 | 95.7 | 65.5 |
> | VPT (shallow) | 78.0 | 92.9 | 72.2 |
> | VPT (deep) | 78.8 | 96.1 | 72.8 |
> | SPT (original paper) | 78.9 | 94.9 | 68.8 |
> | Attention Transfer (shallow) | 81.7 | 96.2 | 77.4 |
> | Attention Transfer (deep) | **81.8** | **96.5** | **77.9** |
>
> We can see that compared with the baselines, attention transfer (deep) achieves a great boost. It also achieves a slight boost compared to its shallow version. But compared with the boost observed from VPT-Shallow to VPT-Deep, a larger performance boost from attention transfer shallow to deep should be expected. Why the boost from attention transfer shallow to deep is not significant enough could be due to semantic mismatch: the source prompt is trained at only the input (the lowest) layer of the language model but applied to every (from the lowest to highest) layer of the vision model. A prompt learned in a lower layer may not be effective on a higher layer, causing the performance boost less significant.
>
> However, we think that with a more sophisticated methodology design, the performance boost can be further enlarged.

---

> ### Author Response · Authors · 2024-11-21
> **Response to weakness 4**
>
> # **Weakness 4 (the accuracy on Natural subset decreases, making it hard to determine the effectiveness. Experiments on more datasets should be conducted to support the claim of this paper)**
>
> Attention transfer underperforms SPT [1] because **SPT uses a better ViT checkpoint than ours**.
>
> We did a systematic check on the baseline SPT using their official code (https://github.com/SinAxCosB/Self-Prompt-Tuning) and found that: SPT uses the ‘vit_base_patch16_224.augreg_in21k’ checkpoint proposed in [2], which is an upgraded version than the original “vit_base_patch16_224_in21k’” checkpoint (which we and the baseline VPT use).
>
> The authors of SPT use timm library to load the pretrained ViT checkpoint named ‘vit_base_patch16_224_in21k’. But in timm, this name actually points to the ‘vit_base_patch16_224.augreg_in21k’ checkpoint (as stated in this link https://huggingface.co/google/vit-base-patch16-224-in21k/discussions/7).
> They probably did not notice that they used a different checkpoint.
>
> We reran SPT using the augreg ViT on some tasks and successfully got what they reported in their paper:
>
> | | ViT Checkpoint | Caltech101 | CIFAR100 | DTD | Flowers |
> | - | - | - | - | - | - |
> | SPT-reported | augreg | 91.2 | 78.9 | 71.2 | 99.4 |
> | SPT-reran | augreg | 90.7 | 80.0 | 72.3 | 99.2 |
>
> We also reran SPT using the original ViT (same to ours):
>
> | | ViT Checkpoint | Caltech101 | CIFAR100 | DTD | Flowers | Pets | Sun397 | SVHN | Patch Camelyon | Resisc45 | EuroSat | Retinopathy | DMLab | KITTI | SmallNorb_azi | SmallNorb_ele | dSprites_loc | dSprites_ori | Clevr_dist | Clevr_count |
> | - | - | - | - | - | - | - | - | - | - | - | - | - | - | - | - | - | - | - | - | - |
> | VPT | original | 88.7 | 78.0 | 65.7 | 97.8 | 87.9 | 50.2 | 71.3 | 80.8 | 75.7 | 92.9 | 74.0 | 39.8 | 72.2 | 17.4 | 22.8 | 72.9 | 30.9 | 57.6 | 50.0 |
> | Attention Transfer (ours) | original | **90.1** | **81.7** | **68.9** | **98.4** | **89.4** | **52.6** | **82.8** | **82.1** | **82.9** | **96.2** | **75.3** | **44.9** | **77.4** | **24.3** | **36.3** | **73.1** | **43.2** | **60.8** | **67.9** |
> | SPT-reran | original | 88.4 | 78.1 | 66.3 | 97.8 | 87.7 | 50.7 | 75.0 | 80.2 | 76.9 | 93.9 | 74.2 | 41.4 | 68.8 | 17.9 | 22.8 | 64.5 | 36.9 | 58.4 | 53.2 |
>
> So the fact is attention transfer is better than SPT on VTAB-1K.
>
> To the best of our knowledge, our investigated cross-modality prompt transfer cooks the best prompt (of length 100) for VPT-Shallow in the entire prompt tuning literature.
>
> Given the information above, we think that it is no longer needed to extend our experiments to more datasets. We hope this could solve your concern! Thank you!
>
> **Note that we haven't added the new results to our paper.**
>
> [1] Wang, Yuzhu, Lechao Cheng, Chaowei Fang, Dingwen Zhang, Manni Duan, and Meng Wang. "Revisiting the power of prompt for visual tuning." arXiv preprint arXiv:2402.02382 (2024).
>
> [2] Steiner, Andreas, Alexander Kolesnikov, Xiaohua Zhai, Ross Wightman, Jakob Uszkoreit, and Lucas Beyer. "How to train your vit? data, augmentation, and regularization in vision transformers." arXiv preprint arXiv:2106.10270 (2021).

---

> ### Author Response · Authors · 2024-11-21
> **Response to weakness 5 and 6**
>
> # **Weakness 5 (Besides, the meaning of designing a good metric estimating prompt transferability should be also clarified. GM & GT outperforms Vdata by 3 points with respect to the mean Kendall's coefficient. To what degree does such an improvement contribute to enhancing the transfer performance on VTAB-1k?)**
>
> For a prompt transferability estimation method, let’s define its top-n accuracy as: the number of target tasks whose best source prompt lies in the top-n transferable source prompts predicted by the method (divided by the total number of target tasks).
>
> For top-1 accuracy, $\mathcal{G}_M \mathcal{G}_T$ is 0.37, while Vdata is 0.32.
>
> For top-2 accuracy, $\mathcal{G}_M \mathcal{G}_T$ is 0.68, while Vdata is 0.63.
>
> For top-3 accuracy, $\mathcal{G}_M \mathcal{G}_T$ is 1, while Vdata is 0.89.
>
> This shows that $\mathcal{G}_M \mathcal{G}_T$ helps a target task better choose the source prompt.
>
> But when it comes to the performance boost: 3% advantage of $\mathcal{G}_M \mathcal{G}_T$ over Vdata (in terms of Kendall's coefficient) doesn’t really make a difference on the transfer performance. Both $\mathcal{G}_M \mathcal{G}_T$ and Vdata are effective in picking out the top-transferable source prompts. And the transfer performance among the top-transferable source prompts does not really differ a lot, as evidenced by the relative performance shown in **Figure 3b.1**.
>
> Vdata is a simple baseline conceptualized by us to show the positive relation between the number of source samples and the transfer performance. It determines transferability purely by the number of source samples, neglecting the relation between source and target tasks. Whereas $\mathcal{G}_M \mathcal{G}_T$ takes full consideration on such relation. We put the comparison between Vdata and $\mathcal{G}_M \mathcal{G}_T$ in our paper to show that capturing the intrinsic connections between tasks is more important than brutally choosing the pretraining dataset based on data volume.
>
> When it comes to the performance boost of $\mathcal{G}_M \mathcal{G}_T$ over the baseline methods (designed for in-modality transfer), it would be considerably larger: If we follow $\mathcal{G}_M \mathcal{G}_T$ choosing the best source prompt it estimates, we will end up with an overall accuracy of 68.2 across all target tasks. But if we use the baseline method ON, the overall accuracy drops to 66.6.
>
>
>
> # **Weakness 6 (discussions about current literature)**
> Thank you deeply for pointing that out! To promote clarity, we have added the discussion in **line 266-269** of our new revision. We also put the discussion here:
>
> *"In the literature, Wang et al. (2023) verified that prompt tuning the query, key, and value in an attention block can help a pretrained model learn new attention patterns required by a target task. Our attention block has similar aims but differs in detail: it also adapts pretrained modules to target tasks through learning new attention patterns. But it only involves the query prompt, as its key and value matrices are not frozen, enabling new attention patterns to be grasped without key and value prompts."*
>
> Overall, our attention transfer and APrompt share a similar goal: adapting a set of pretrained weights to a target task by enabling an attention block to grasp new attention patterns. Whereas the difference between our attention transfer and APrompt lies in whether the attention block is frozen or not. APrompt deals with frozen attention blocks and introduces prompts to query, key, and value matrices to enable the attention block to learn new patterns. While our attention transfer deals with a trainable attention block. Therefore, it can capture new patterns by only using the query prompt.
>
> We hope the above discussion answers your question. Thank you!
>
> Wang, Qifan, Yuning Mao, Jingang Wang, Hanchao Yu, Shaoliang Nie, Sinong Wang, Fuli Feng et al. "Aprompt: Attention prompt tuning for efficient adaptation of pre-trained language models." In Proceedings of the 2023 Conference on Empirical Methods in Natural Language Processing, pp. 9147-9160. 2023.

---

> > ### Comment · Reviewer_Ve1v · 2024-11-25
> > **My concerns are addressed.**
> >
> > Thank you for your response. My concerns are addressed, I am willing to increase my score.

---

> > > ### Author Response · Authors · 2024-11-25
> > > **Appreciation for Your Feedback and Support**
> > >
> > > Dear Reviewer Ve1v,
> > >
> > > It's glad to know that our responses address your concern.
> > > Thank you deeply for your valuable feedback and for indicating your willingness to consider a higher score for our paper.
> > > We hereby express our deepest thanks to your thoughtful evaluation and the time you’ve spent on our submission.
> > >
> > > We noted that the system does not yet show the score adjustment you mentioned in your reply. If it’s still under consideration, we completely understand and are happy to provide any additional clarification or information that might help mitigate your concern.
> > >
> > > Thank you once again for your support and feedback! They are very helpful to the improvement of our paper!
> > >
> > > Best regards,
> > >
> > > Authors of Submission 2711

---

> ### Comment · Area_Chair_4uiP · 2024-11-25
>
> Dear Reviewer Ve1v,
>
> Could you kindly review the rebuttal thoroughly and let us know whether the authors have adequately addressed the issues raised or if you have any further questions.
>
> Best,
>
> AC of Submission2711

---

### Official Review · Reviewer_v8xH · 2024-11-03

**Soundness:** 3
**Presentation:** 3
**Contribution:** 3
**Rating:** 6
**Confidence:** 3

**Summary:**

- Prompt tuning has become a popular paradigm to adapt frozen models to new tasks.
- While prompt-transfer within the same modality has shown to been helpful, the paper argues that this area has not been thoroughly explored.
- The paper presents an analysis on feasibility of cross-model prompt transfer which is especially relevant for data scarce modalities. The paper studies various design choices through experiments and analysis.
- L205-208: To do so, the problem is split into modality and task-based gaps and these are quantified. A new approach called attention transfer is introduced to reduce this gap which helps in improving the benchmarks. One should not have to go to the appendix to understand the intuitions behind the approach.

**Strengths:**

- The paper proposes an interesting idea of transferring language prompts to vision for various data-scarce tasks. The idea is interesting and seems novel.
- The paper is easy to follow for the most part.
- Experiments are thorough. Many different ablations are presented, various design choices are explored.
- The results seem promising and show benefit of using the approach.

**Weaknesses:**

While the overall idea, experiments and findings could be helpful to the community, I have the following concerns with this paper:

- It is unclear why you want to quantify prompt transferability (equation 6) given it is a function of hyperparameters, training data etc. It would be helpful to provide some intuitions.
- L274: What was the motivation behind chosing a IN-21k pre-trained checkpoint chosen ? Why not 1k ? Why not a language aligned model like CLIP ?
- Intuitions: Maybe I missed this in the paper, but a discussion on why should prompts be transferrable, what is the composition of the dataset where they transfer and where they don't will be very helpful to the reader.

**Questions:**

Clarity:
- L106-107: Unclear from context.
- The discussion on the hypothesis on why cross-modal prompt transfer should even work must be discussed early to set stage for the problem and the proposed solution
- Is this a key contribution of the paper - if so this must be discussed in the main paper rather than the appendix.
- L269: wouldn't eq (8) increase the prompt size to > 100 ?
- How is "relative performance" defined? Given how dense the figure is, authors must include conclusion of this experiment in the caption.
- Statistical significance: Are the improvements statistically significant ? I see that the standard deviations are reported in the appendix. Authors should include a discussion on significance of improvements in the main paper.

---

> ### Author Response · Authors · 2024-11-21
> **Response to weaknesses**
>
> # **Weakness 1 (why equation 6 is a function of hyperparameters and training data)**
>
> Thank you deeply for your valuable suggestions!
>
> We will answer this weakness from two aspects:
>
> The first is our experiment results: when equation 6 is not a function of hyperparameters and training data (i.e. remove the universal projector $P_u$ in equation 5 and directly calculate the cosine similarity between source and target prompts as task gap), we can only achieve very poor estimation of the transfer results, as evidenced by the Avg Cos row in Table 2.
> Whereas the $\mathcal{G}_T$ Only row in Table 2 shows the results of calculating the cosine similarity between a target prompt and the source prompts projected by the universal projector $P_u$. It’s clear that with the use of the universal projector $P_u$, we can estimate the prompt transferability much reliably.
>
> The second is the necessity of the universal projector: The source prompts are trained on NLP tasks with a language model, whereas the target prompts are trained on CV tasks with a vision model, making the source and target prompts share significant differences that make any direct similarity calculations less informative. Our universal projector is a simple trick to mitigate this difference by casting the source prompts to the target prompt space, thereby making similarity calculations between source and target prompts feasible and meaningful.
>
>
>
> # **Weakness 2 (why IN-21k pre-trained checkpoint)**
>
> We choose the ImageNet-21k ViT checkpoint because we want to ensure fair comparisons between our method and the baseline methods.
>
> We want to see how much can cross-modality prompt transfer boost Visual Prompt Tuning (VPT) [1], so it’s important to use the same checkpoint as VPT does. And VPT uses the ViT checkpoint pretrained on ImageNet-21k.
>
> Currently we focus on the prompt transfer between models pretrained on a single modality. We believe that our current results are enough to support our conclusions. Hope this can solve your concern, thank you!
>
> [1] Jia, Menglin, Luming Tang, Bor-Chun Chen, Claire Cardie, Serge Belongie, Bharath Hariharan, and Ser-Nam Lim. "Visual prompt tuning." In European Conference on Computer Vision, pp. 709-727. Cham: Springer Nature Switzerland, 2022.
>
>
>
> # **Weakness 3 (why are the prompts transferrable; the composition of the dataset where they transfer and where they don't)**
>
> Regarding why the prompts are transferrable, thank you for pointing out. We agree with your suggestion, and we have added a Discussion section in our **Appendix E**, briefly discussed about why the prompts are transferrable. We also put our discussions here:
>
> *"(i) Transferable source prompts are typically pretrained on a huge volume of NLP data. As a result, these prompts should contain valuable semantics that are high-level enough to be extendable to other modalities. Moreover, they should also have a generalisability powerful enough to generalize to an unseen modality.
> (ii) On the other hand, language models trained on large and diverse datasets, learn representations that capture concepts, relationships, and structures beyond text alone. These abstract features may generalize well to other modalities, where the prompts can act as conceptual "anchors" that “carry” the representations across modalities."*
>
> Regarding what would a transferable or less-transferable target task looks like, we discussed this in line 411 of our initial submission (**line 407 in our new uploaded revision**): target tasks that already benefit a lot from prompt tuning would be more transferable (i.e. benefit even more from transferring the NLP-pretrained prompts) and vice versa.
> Here we provide a brief discussion about this conclusion (which we have also put into our Discussion section in **Appendix E**):
>
> *"If a target task benefits a lot from prompt tuning (i.e. has a higher prompt tuning performance than linear probing), this indicates that this target task requires the pretrained model to learn new features, because simple linear probing on its original features does not help much.
> In another word, such a target task would require the pretrained model to learn extra knowledge. Therefore, why cross-modality prompt transfer excels on such a target task is easy to understand: it introduces more knowledge to the target task than vanilla prompt tuning. As vanilla prompt tuning only learns target knowledge while cross-modality prompt transfer learns both the source and target knowledge.
> In conclusion, if a target task requires the pretrained model to grasp a lot of new features, then it would be a more transferable target task, and vice versa."*

---

> ### Author Response · Authors · 2024-11-21
> **Response to questions 1-5**
>
> # **Question 1(L106-107: Unclear from context)**
> Thank you for pointing this out, we have now modified the sentence, it is now in **line 123-126** in our new revision:
>
> *"Our investigated cross-modality prompt transfer can be regarded as initializing the prompt with adapted NLP-pretrained prompt. It and SPT both focus on prompt initialization but are from different perspectives. Therefore, to demonstrate the effectiveness of cross-modality prompt transfer, SPT is chosen for comparison."*
>
> What we want to express is: cross-modality prompt transfer, although we use different methods to adapt the source prompt, is still a method focusing on how to better initialize the prompt vectors for a target task. In this case, the best way to demonstrate its effectiveness is through comparisons with other methods focusing on prompt initialization. SPT suits this case perfectly and it’s a newly released baseline published in ICML2024. Considering its timeliness and shared focus, we include SPT for comparison.
>
>
> # **Question 2 (The discussion on the hypothesis on why cross-modal prompt transfer should even work must be discussed early to set stage for the problem and the proposed solution)**
>
> Thank you for your suggestion. We have added the statement of the hypothesis in our introduction section, you can find it in **line 53-54** in our new revision:
>
> *“In the literature, prompt transfer has been proven effective between NLP tasks (Vu et al., 2022; Su et al., 2022). Based on the findings, we hypothesize that such practice can be safely extended to cross-modality scenarios and conduct extensive studies to verify our hypothesis.”*
>
> This hypothesis is fully built upon previous research that focuses on transferring pretrained prompts across different NLP tasks, on which the pretrained prompts demonstrate astonishing zero-shot and prompt tuning performance. It would be natural to make such hypothesis that the NLP-pretrained prompts could also be effective on other modality tasks. One aim of our work is to verify the feasibility of cross-modality prompt transfer.
>
> # **Question 3 (Is this a key contribution of the paper - if so, this must be discussed in the main paper rather than the appendix)**
>
> We have a little bit of trouble locating which part of appendix you are referring to. Some key information might have been lost in your comments when the reviews are made visible to us. If there are some parts needed to be discussed in the main paper, please let us know.
>
> Our key contributions were listed in the introduction section of our main paper, covering three aspects: (i) verifying the feasibility of cross-modality prompt transfer; (ii) designing a transferability estimation metric; and (iii) proposing attention transfer to further boost the performance. We think that all three of our contributions were detailed and verified in our main paper. The appendix is mainly there to provide explanatory and supplementary information for our contributions. Thank you!
>
>
>
> # **Question 4 (wouldn't eq (8) increase the prompt size to > 100?)**
>
> Equation 8 will not increase the prompt size to greater than 100.
>
> We carefully design the length of the adapted source prompt ($C(p_s)$) and target prompt ($p_t$), to make them have an overall length of 100.
>
> Our proposed Prompt Concentrator (C) can condense the source prompt from length 100 to any desired length. For example, we can condense the source prompt from length 100 to 50 with the Prompt Concentrator, and concatenate 50 target prompts to the adapted (condensed) source prompt, making the final prompt length to be 100.
>
>
> # **Question 5 (How is "relative performance" defined?)**
>
> Thank you for your suggestion. To avoid confusion, we have added the explanation of the relative performance in both **Figure 3 and Table 1**.
>
> **Figure 3**:
>
> *“The relative performance is calculated by (Performance − B)/B × 100%, where B is the performance of transferring a random prompt, conducting vanilla linear probing or VPT, depending on the situations stated in the figure.”*
>
> **Table 1**:
>
> *“V/L = (VPT − Linear)/Linear × 100% shows the relative gain of VPT over Linear.
> P/V = (Proj. − VPT)/VPT × 100% shows the relative gain of Proj. over VPT.”*

---

> ### Author Response · Authors · 2024-11-21
> **Response to question 6**
>
> # **Question 6 (Statistical significance: Are the improvements statistically significant? I see that the standard deviations are reported in the appendix. Authors should include a discussion on significance of improvements in the main paper)**
>
> Thank you for your suggestion. The improvements are statistically significant.
>
> We prove the statistical significance of the improvements following the below steps:
>
> Step 1: Define our null hypothesis: the improvements are not introduced by transferring the source prompts.
>
> Step 2: We choose one-sided test as the effects will be in one direction (i.e. contribute to the increase of the improvements)
>
> Step 3: Our data
>
> Group 1 (Conducting projection transfer on all 19 target tasks with a randomly initialized prompt): In this case, we have only one “source” prompt and 19 target tasks, which makes the total number of observations to be 19. Among these 19 observations, only 5 show improvements over vanilla prompt tuning, resulting in a percentage of 26.32%. (The data comes from the first row of Figure 3b.2)
>
> Group 2 (Conducting projection transfer on all 19 target tasks with all 13 source tasks): In this case, we have 13 source prompts and 19 target tasks, resulting in a total number of 19*13=247 observations. Among these 247 observations, 158 show improvements over vanilla prompt tuning, with a percentage of 63.97%. (The data comes from the remaining rows of Figure 3b.2)
>
> Step 4: p-value
> The final p-value is calculated to be 0.02%. That is, given the results we observed, there’s only a chance of 0.02% that the improvements are not introduced by cross-modality prompt transfer. In another word, we are 99.98% certain that the improvements are statistical significant: they are caused by cross-modality prompt transfer, instead of error or random chances.
>
> Plus, our results in **Figure 3a.2** also suggest that introducing cross-modality prompt transfer can help improve the performance on target tasks. In conclusion, we believe that our observations are enough to support our claim: cross-modality prompt transfer improves the prompt tuning performance on tasks in another modality. Thank you!
>
> We have briefly discussed about the statistical significance of the improvements in **line 393-394** of our new revision:
>
> *"Additionally, the improvements observed in Figure 3b.2 are proven to be statistically significant: they are caused by transferring the source prompts, instead of error or random chances (the prove can be found in our appendix)."*

---

> ### Comment · Reviewer_v8xH · 2024-11-25
> **Thanks for the detailed comments**
>
> Thanks for the detailed comments. They were very helpful.
>
> - Weakness 1 (why equation 6 is a function of hyperparameters and training data): Empirical evidence of using the projector is helpful for understanding. My concern was because of how the gap now depends on well the projector is trained, but a universal $P_u$ helps avoid this problem to an extent.
> - Weakness 2 (IN-21k) : I agree. While a fair comparison to prior work is essential (which is from 2022), a discussion on why a image-based model vs language-aligned model (like CLIP) will help the reader especially given the problem being tackled.
>
> Since my score is at 6 already (and I am not an expert in this area), I will vote to accept the paper and am happy to increase the score after discussing with other reviewers.

---

> > ### Author Response · Authors · 2024-11-25
> > **Appreciation for Your Feedback and Support**
> >
> > Dear Reviewer v8xH,
> >
> > Thank you very much for providing your thoughtful feedback and for spending your valuable time on our work. We are glad to know that our responses were helpful!
> >
> > + **Weakness 1:** We appreciate your acknowledgment of the empirical evidence and your perspective on how a universal projector can mitigate the issue in cross-modality prompt transfer. Your insights are valuable for improving the clarity of our paper.
> >
> > + **Weakness 2:** We completely agree that a discussion on why we choose an image-based model over a language-aligned model (like CLIP) would be beneficial for readers. Below is our discussion about our choice:
> >
> > *"We discuss our choice on the supervised ImageNet-21k ViT checkpoint (ViT-IN21k for short) from two perspectives:
> > (i) **Why not a checkpoint pretrained on a different dataset or by a different pretraining objective**: we need to demonstrate how significantly can cross-modality prompt transfer boost prompt tuning. In the meantime, visual prompt tuning was initially experimented on the ViT-IN21k checkpoint. Therefore, to make the comparison meaningful, we select the ViT-IN21k checkpoint.
> > (ii) **Why not a language-aligned ViT checkpoint**: we need to first demonstrate that a model, even pretrained without any textual knowledge (e.g. the ViT-IN21k checkpoint), can benefit from transferring prompts pretrained on pure textual data. Second, the source prompts are trained with language models on language tasks. The language models are not aligned with any variations of ViT checkpoint and the language tasks are not aligned with any image tasks. We believe that a language-aligned ViT checkpoint should be applied to a situation where there are connections across modalities. However in our cases, the two modalities are isolated. Therefore, we choose the ViT-IN21k checkpoint instead of a language-aligned ViT checkpoint."*
> >
> > If our paper gets accepted, we will put this discussion in our Appendix E. We hope the above discussion can answer your concern!
> >
> > At last, we are deeply grateful for your willingness to support the acceptance of our paper and for considering a score adjustment. Your feedback and encouragement mean a lot to us, and they have been of great help in refining our research.
> >
> > Thank you again for your thoughtful engagement and support. Please don’t hesitate to let us know if there’s anything else we can clarify or improve.
> >
> > Best regards,
> >
> > Authors of Submission 2711

---

### Official Review · Reviewer_mrx7 · 2024-11-05

**Soundness:** 3
**Presentation:** 3
**Contribution:** 3
**Rating:** 6
**Confidence:** 5

**Summary:**

This paper explores cross-modality prompt transfer, aiming to improve the performance of prompt-tuning methods when transferring prompts across different data modalities, specifically from NLP to CV tasks. The authors propose a novel approach using attention transfer and modality gap analysis to address data scarcity challenges in the target domain. Experiments on a wide variety of tasks demonstrate the feasibility and advantages of this cross-modality prompt transfer, highlighting the potential of leveraging rich source prompts for enhancing performance in data-limited scenarios.

**Strengths:**

- The paper is well-organized and clearly written.
- The paper introduces a new perspective in prompt transfer by exploring cross-modal prompt adaptation
- The authors conduct a thorough analysis of the modality and task gaps, which improves the understanding of transferability in cross-modal scenarios, offering valuable insights for future research .
- Attention transfer is proposed to mitigate modality and task gaps, offering an effective way to reuse and adapt prompts across modalities.
- The paper provides substantial experimental evidence on multiple datasets, showing that cross-modal prompt transfer can yield performance comparable to, or even surpassing, recent prompt-based benchmarks .

**Weaknesses:**

- The paper would benefit from additional comparisons with same-modality prompt transfer baselines, which would help quantify the specific impact of cross-modal transfer approaches.
- The approach seems sensitive to hyperparameters, which could affect reproducibility and performance stability, particularly in diverse target environments.
- There are instances where cross-modal transfer underperforms or fails, the approach's limitations should be further discussed.

**Questions:**

Although the paper provides theoretical and experimental support, practical applications, particularly real-world tasks where cross-modality transfer excels, are not thoroughly discussed. Besides, the method may face scalability issues when applied to larger models or when requiring complex attention transfer setups, which could limit its applicability in more resource-intensive settings .

---

> ### Author Response · Authors · 2024-11-21
> **Response to weakness 1 and 2**
>
> # **Weakness 1 (The paper would benefit from additional comparisons with same-modality prompt transfer baselines, which would help quantify the specific impact of cross-modal transfer approaches)**
>
> We express our deepest gratitude for your valuable suggestions!
>
> Currently, to the best of our knowledge, we could not find a same-modality prompt transfer baseline designed for CV tasks (that is, a prompt transfer method that has a source task modality of image and a target task modality of image). However, there are several works focused on same-modality prompt transfer between NLP tasks, but most of them focus on fusing multiple source prompts for a target NLP task [1] [2] [3]. Their setup differs from ours, as shown in the following table.
>
> | | Source Task Modality | Number of Source Tasks | Target Task Modality |
> | - | - | - | - |
> | [1] | Text | >1 | Text |
> | [2] | Text | >1 | Text |
> | [3] | Text | >1 | Text |
> | VPT | / | 0 | Image |
> | SPT | / | 0 | Image |
> | Cross-modality prompt transfer | Text | 1 | Image |
>
> We can see that methods proposed in [1][2][3] differ greatly from our setup. It would be unfair and less-effective to involve them in comparison, if we wish to explore the specific impact of cross-modality prompt transfer.
>
> The baseline methods (VPT and SPT), on the other hand, perfectly ablate the part whose effectiveness we want to verify: their setup differs from cross-modality prompt transfer on whether the prompt is pretrained on a single source task. This is also the reason why they are chosen as our baselines: comparing with VPT helps to examine whether pretraining the source prompt on a source task would contribute to performance boost on a different modality task; while comparing with SPT helps to demonstrate the effectiveness of our proposed cross-modality prompt transfer. We believe these comparisons are enough to quantify the specific impact of cross-modality prompt transfer. Thank you!
>
> [1] Asai, Akari, Mohammadreza Salehi, Matthew E. Peters, and Hannaneh Hajishirzi. "ATTEMPT: Parameter-efficient multi-task tuning via attentional mixtures of soft prompts." arXiv preprint arXiv:2205.11961 (2022).
>
> [2] Lv, Xingtai, Ning Ding, Yujia Qin, Zhiyuan Liu, and Maosong Sun. "Parameter-efficient weight ensembling facilitates task-level knowledge transfer." In Proceedings of the 61st Annual Meeting of the Association for Computational Linguistics (Volume 2: Short Papers), pp. 270-282. 2023.
>
> [3] Wang, Zhen, Rameswar Panda, Leonid Karlinsky, Rogerio Feris, Huan Sun, and Yoon Kim. "Multitask prompt tuning enables parameter-efficient transfer learning." arXiv preprint arXiv:2303.02861 (2023).
>
>
> # **Weakness 2 (The approach seems sensitive to hyperparameters, which could affect reproducibility and performance stability, particularly in diverse target environments)**
>
> Thank you for your suggestion.
>
> In fact, the results of our projection transfer were obtained with the same set of hyperparameters across all source-target transfer pairs. Given the fact that projection transfer obtains much better performance than VPT (which performed hyperparameter grid-search on every target task), we think that the optimization of projection transfer is not very susceptible to hyperparameters. As a result, projection transfer will not be limited much in diverse target environments.
>
> As for attention transfer, it introduces more trainable parameters than projection transfer, making us spend a grid-search step to settle its hyperparameters. But the hyperparameter search grid is considerably smaller than the search grid of baseline methods (attention transfer has a search grid of size 32 consisting of 8 learning rate values and 4 weight decay values, while the baseline VPT has a search grid of 44, as reported in its original paper), as we find that the optimal hyperparameters of attention transfer are similar across different target tasks (with a typical learning rate value ranged from 0.001 to 0.01, and a most used weight decay value of 0.001). We believe that this similarity of hyperparameters can help attention transfer function under a diverse target environment. For example, on Caltech101, CIFAR100, DTD, and Kitti, attention transfer uses the exact same set of hyperparameters, enabling it to be trained simultaneously on these tasks. Therefore, attention transfer will not be influenced much in a diverse target environment, as it does not heavily rely on hyperparameters.
>
> As for our reproducibility, we have provided our code and our training hyperparameters in our appendix, we are confident that our results are reproducible using the resources we provided. Thank you!

---

> ### Author Response · Authors · 2024-11-21
> **Response to weakness 3**
>
> # **Weakness 3 (There are instances where cross-modal transfer underperforms or fails, the approach's limitations should be further discussed)**
>
> Cross-modality prompt transfer underperforms the baseline SPT on some target tasks because SPT uses a better ViT pretrained checkpoint.
>
> We did a systematic check on the baseline SPT using their official code (https://github.com/SinAxCosB/Self-Prompt-Tuning) and found that: SPT uses the ‘vit_base_patch16_224.augreg_in21k’ checkpoint proposed in *“Steiner, Andreas, Alexander Kolesnikov, Xiaohua Zhai, Ross Wightman, Jakob Uszkoreit, and Lucas Beyer. "How to train your vit? data, augmentation, and regularization in vision transformers." arXiv preprint arXiv:2106.10270 (2021).”*, which is an upgraded version than the original “vit_base_patch16_224_in21k’” checkpoint (which we and the baseline VPT use).
>
> The authors of SPT use timm library to load the pretrained ViT checkpoint named ‘vit_base_patch16_224_in21k’. But in timm, this name actually points to the ‘vit_base_patch16_224.augreg_in21k’ checkpoint (as stated in this link https://huggingface.co/google/vit-base-patch16-224-in21k/discussions/7).
> They probably did not notice that they used a different checkpoint.
>
> We reran SPT using the augreg ViT on some tasks and successfully got what they reported in their paper:
> | | ViT Checkpoint | Caltech101 | CIFAR100 | DTD | Flowers |
> | - | - | - | - | - | - |
> | SPT-reported | augreg | 91.2 | 78.9 | 71.2 | 99.4 |
> | SPT-reran | augreg | 90.7 | 80.0 | 72.3 | 99.2 |
>
> We also reran SPT using the original ViT (same to ours):
>
> | | ViT Checkpoint | Caltech101 | CIFAR100 | DTD | Flowers | Pets | Sun397 | SVHN | Patch Camelyon | Resisc45 | EuroSat | Retinopathy | DMLab | KITTI | SmallNorb_azi | SmallNorb_ele | dSprites_loc | dSprites_ori | Clevr_dist | Clevr_count |
> | - | - | - | - | - | - | - | - | - | - | - | - | - | - | - | - | - | - | - | - | - |
> | VPT | original | 88.7 | 78.0 | 65.7 | 97.8 | 87.9 | 50.2 | 71.3 | 80.8 | 75.7 | 92.9 | 74.0 | 39.8 | 72.2 | 17.4 | 22.8 | 72.9 | 30.9 | 57.6 | 50.0 |
> | Attention Transfer (ours) | original | **90.1** | **81.7** | **68.9** | **98.4** | **89.4** | **52.6** | **82.8** | **82.1** | **82.9** | **96.2** | **75.3** | **44.9** | **77.4** | **24.3** | **36.3** | **73.1** | **43.2** | **60.8** | **67.9** |
> | SPT-reran | original | 88.4 | 78.1 | 66.3 | 97.8 | 87.7 | 50.7 | 75.0 | 80.2 | 76.9 | 93.9 | 74.2 | 41.4 | 68.8 | 17.9 | 22.8 | 64.5 | 36.9 | 58.4 | 53.2 |
>
> Therefore, the fact shall be: cross-modality prompt transfer is better than SPT on VTAB-1K. We hope this could address your concern. Thank you!
>
> **Note that we haven't added the new results to our paper.**

---

> ### Author Response · Authors · 2024-11-21
> **Response to questions**
>
> # **Question 1 (Although the paper provides theoretical and experimental support, practical applications, particularly real-world tasks where cross-modality transfer excels, are not thoroughly discussed)**
>
> Thank you for your suggestion! We did mention a few about cross-modality prompt transfer can help mitigate challenges faced by data-scarce modalities in our introduction. However, due to page limitations, we have added a more in-depth discussion about the real-world scenarios in our **Appendix E**. The discussion is also pasted below:
>
> *"In terms of real-world scenarios, cross-modality prompt transfer can be of help from two aspects:
> (i) improve the prompt tuning performance on data-scarce tasks or privacy sensitive tasks (for example, medical diagnosis or business analysis), as it transfers abundant source knowledge without depending on the source data.
> (ii) improve the prompt tuning performance on tasks that would benefit from combining knowledge across text and vision domains, especially when one modality has abundant labelled data or provides complementary insights. For example, it can enhance medical image analysis by transferring prompts pretrained on text-based medical knowledge, support vision systems used in autonomous vehicles by transferring prompts trained on traffic rule texts, etc.*
>
>
> # **Question 2 (Besides, the method may face scalability issues when applied to larger models or when requiring complex attention transfer setups, which could limit its applicability in more resource-intensive settings)"**
>
> Thank you for pointing this out.
>
> In fact, our projection transfer and attention transfer can be easily scaled to different scenarios:
> + The source language model is smaller than the target vision model (i.e. the trained source prompts have a smaller dimension than the target vision model’s hidden dimension).
> + The source language model is larger than the target vision model (i.e. the trained source prompts have a larger dimension than the target vision model’s hidden dimension).
>
> We show that projection transfer and attention transfer can be extended to these scenarios in a really simple way:
>
> Assume the source prompts have a dimension of $d_s$ and the target model requires its input to have a dimension of $d_t$.
>
> For projection transfer, since the prompt projector is a linear layer, we can solve this dimension mismatch simply by making the projection matrix to have a dimension of $d_s×d_t$.
>
> For attention transfer, this dimension mismatch can be solved by: (i) making the query vectors to have a dimension of $d_t$; (ii) making the key and value projection matrices to have a dimension of $d_s×d_t$.
>
> The scalability is an important factor we consider when we design our method. Therefore, our projection transfer and attention transfer have the ability to be extended to larger or even smaller models.
>
> As for setups that require complex attention transfer: if we need a very complex attention transfer to adapt a source prompt, we’d say that perhaps this source prompt is not an ideal one and we should find a more suitable source prompt. In all our scenarios, the source prompts are adapted by a linear projector (both in projection transfer and attention transfer) and showed lots of positive results. Therefore, we think that we do not need very complex operations to adapt a source prompt.

---

> ### Comment · Area_Chair_4uiP · 2024-11-25
>
> Dear Reviewer mrx7,
>
> Could you kindly review the rebuttal thoroughly and let us know whether the authors have adequately addressed the issues raised or if you have any further questions.
>
> Best,
>
> AC of Submission2711

---

> > ### Comment · Reviewer_mrx7 · 2024-11-26
> > **Most of my concerns are addressed in the response.**
> >
> > Thanks for the authors' response, most of my concerns are well addressed, I decide to keep my score.

---

> > > ### Author Response · Authors · 2024-12-02
> > > **Thank you for your comments**
> > >
> > > Thank you very much for your thoughtful feedback on our response. We are glad to hear that most of your concerns have been addressed. Your insights have been instrumental in helping us refine and improve the clarity of our work.
> > >
> > > If there are any remaining areas where additional clarification or further evidence would be helpful, please do let us know—we are more than happy to provide additional details to strengthen our submission further.
> > >
> > > Best wishes,
> > >
> > > Authors of submission 2711

---

### Official Review · Reviewer_4VnN · 2024-11-06

**Soundness:** 3
**Presentation:** 3
**Contribution:** 2
**Rating:** 6
**Confidence:** 4

**Summary:**

This paper explores several aspects of prompt tuning in cross-modality settings (e.g., vision and language), focusing on feasibility, transferability estimation, and performance improvement. Specifically, the authors validate that source prompts pretrained on text data can be safely transferred to tasks in a different modality. They also propose a better metric for transferability estimation, which takes into account both modality and task gaps. Finally, the paper introduces attention transfer techniques for cross-modality prompt transfer, which are shown to perform comparably to single-modality prompt tuning algorithms such as SPT.

**Strengths:**

* The motivation is good.
* The overall logic and flow of the paper are clear.
* The experiments are thorough, with a detailed exploration of various aspects of cross-modality prompt tuning.

**Weaknesses:**

* Although the paper claims that cross-modality prompt transfer can boost performance in data-scarce tasks, this is not convincingly demonstrated in the experiments, particularly when comparing to the SPT baseline.
* The problem itself is interesting, and some findings, such as the proposed transferability estimation metric and the connection between single-modality and cross-modality prompt tuning, are valuable. However, the advantage of cross-modality prompt tuning over single-modality methods is not convincingly established.
* The proposed transferability metric seems somewhat complex. It requires training a universal projector across tasks, which may introduce additional training costs and overhead.
* The experiments lack strong evidence to show a clear advantage of cross-modality transfer, especially in data-scarce settings.

**Questions:**

1. What is the training cost associated with the proposed transferability metric? How does it scale with the number of tasks or modalities?
2. What is the distribution of modality and task gaps in practice? Is there a recommended threshold to select the source prompt for transfer, instead of choosing the best?
3. Could the paper provide more concrete evidence of improvements in data-scarce tasks using the proposed cross-modality transfer approach?
4. Is there a way to make the transferability metric more efficient, reducing the need for the universal projector?

---

> ### Author Response · Authors · 2024-11-21
> **Response to weakness 1**
>
> Great thanks for your valuable suggestions!
>
> Regarding **weakness 1** (cross-modality prompt transfer falls below SPT):
>
> In search of the why, we did a systematic check on the baseline SPT using their official code (https://github.com/SinAxCosB/Self-Prompt-Tuning) and found that: **SPT uses a better ViT pretrained checkpoint compared to ours**.
>
> The supervised ViT checkpoint SPT uses is the augreg version proposed in *“Steiner, Andreas, Alexander Kolesnikov, Xiaohua Zhai, Ross Wightman, Jakob Uszkoreit, and Lucas Beyer. "How to train your vit? data, augmentation, and regularization in vision transformers." arXiv preprint arXiv:2106.10270 (2021).”*
> Which is an upgraded version than the original ViT. The authors of SPT use timm library to load the pretrained ViT checkpoint named ‘vit_base_patch16_224_in21k’. But in timm, this name actually points to the ‘vit_base_patch16_224.augreg_in21k’ checkpoint (as stated in this link https://huggingface.co/google/vit-base-patch16-224-in21k/discussions/7).
>
> Whereas the ViT checkpoint we and VPT use is the original one. The disparity between model checkpoints introduces the major differences between SPT and our cross-modality prompt transfer.
> We reran SPT using the augreg ViT on some tasks and successfully got what was reported in SPT’s paper:
>
> | | ViT Checkpoint | Caltech101 | CIFAR100 | DTD | Flowers |
> | - | - | - | - | - | - |
> | SPT-reported | augreg | 91.2 | 78.9 | 71.2 | 99.4 |
> | SPT-reran | augreg | 90.7 | 80.0 | 72.3 | 99.2 |
>
> We also reran SPT using the original ViT (same to ours):
>
> | | ViT Checkpoint | Caltech101 | CIFAR100 | DTD | Flowers | Pets | Sun397 | SVHN | Patch Camelyon | Resisc45 | EuroSat | Retinopathy | DMLab | KITTI | SmallNorb_azi | SmallNorb_ele | dSprites_loc | dSprites_ori | Clevr_dist | Clevr_count |
> | - | - | - | - | - | - | - | - | - | - | - | - | - | - | - | - | - | - | - | - | - |
> | VPT | original | 88.7 | 78.0 | 65.7 | 97.8 | 87.9 | 50.2 | 71.3 | 80.8 | 75.7 | 92.9 | 74.0 | 39.8 | 72.2 | 17.4 | 22.8 | 72.9 | 30.9 | 57.6 | 50.0 |
> | Attention Transfer (ours) | original | **90.1** | **81.7** | **68.9** | **98.4** | **89.4** | **52.6** | **82.8** | **82.1** | **82.9** | **96.2** | **75.3** | **44.9** | **77.4** | **24.3** | **36.3** | **73.1** | **43.2** | **60.8** | **67.9** |
> | SPT-reran | original | 88.4 | 78.1 | 66.3 | 97.8 | 87.7 | 50.7 | 75.0 | 80.2 | 76.9 | 93.9 | 74.2 | 41.4 | 68.8 | 17.9 | 22.8 | 64.5 | 36.9 | 58.4 | 53.2 |
>
> So the truth is if SPT uses the same ViT checkpoint to ours, it cannot match cross-modality prompt transfer.
> To the best of our knowledge, our investigated cross-modality prompt transfer cooks the best prompt (of length 100) for VPT-Shallow in the entire prompt tuning literature.
> We hope this could solve your concern! Thank you!
>
> **Note that we haven't added the new results to our paper.**

---

> > ### Author Response · Authors · 2024-11-21
> > **Response to weakness 2-4**
> >
> > # **Weaknesses 2 (the advantage of cross-modality prompt is not convincingly established)**
> >
> > We answer this weakness from two different perspectives:
> >
> > One is from our experimental results, with our new discovery about SPT (as shown in weakness 1), we think that the advantage of cross-modality prompt tuning shall be clearly demonstrated.
> >
> > The second is from our motivation, we highlight the importance and necessity of cross-modality prompt transfer: it aims to benefit some data-scarce modalities who do not have a large dataset to pretrain a prompt.
> >
> > Yes, if we pretrain the prompt on an image task and transfer the prompt to a new image task, the performance would be better than transferring an NLP prompt. But not all modalities would have a dataset suitable for pretraining the prompt (for example, health care data or privacy-sensitive data). In such a scenario, cross-modality prompt transfer has the advantage over tuning based methods in boosting the performance by transferring a knowledge-rich prompt pretrained on a data-rich modality.
> >
> > Our research verifies that feasibility and provides some simple tricks for picking and adapting the NLP-pretrained prompts, by conducting experiments on NLP-to-CV transfer tasks. As we believe text and image have enough big discrepancies, and if this path is clear, then various modalities can be empowered.
> >
> >
> >
> > # **Weakness 3 (the proposed transferability metric being complex)**
> >
> > Thank you for your suggestion, the metric is somewhat complex as it introduces a small number of learnable weights. However, given our current understanding and experimental results, introducing this complexity to our transferability metric is necessary to ensure that we can pick the top-transferable source prompts more reliably.
> >
> > We follow previous works [1] and [2] treating trained prompts as task embeddings and measuring prompt similarities as task gaps. However, in cross-modality scenarios, source prompts and target prompts lie in different semantic spaces. Without putting them in a unified space, brutal-force similarity calculations just make less sense. Therefore, we conceptualized this universal projector.
> >
> > In our future work, we plan to jump outside of the box of “treating trained prompts as task embeddings” and make our efforts in developing less-complex and more-accurate transferability metrics.
> >
> > We will further talk about its complexity and necessity in our response to your questions 1, 2, and 4. Thank you!
> >
> >
> >
> > # **Weaknesses 4 (lack strong evidence to show a clear advantage of cross-modality transfer)**
> >
> > After looking deeper into the results provided by the original VPT paper [3], we have some new supports for demonstrating the advantage of cross-modality transfer. More details about the supports can be found in our response to your question 3, as it is related to this weakness. Thank you!
> >
> >
> > [1] Vu, Tu, Brian Lester, Noah Constant, Rami Al-Rfou, and Daniel Cer. "Spot: Better frozen model adaptation through soft prompt transfer." arXiv preprint arXiv:2110.07904 (2021).
> >
> > [2] Su, Yusheng, Xiaozhi Wang, Yujia Qin, Chi-Min Chan, Yankai Lin, Huadong Wang, Kaiyue Wen et al. "On transferability of prompt tuning for natural language processing." arXiv preprint arXiv:2111.06719 (2021).
> >
> > [3] Jia, Menglin, Luming Tang, Bor-Chun Chen, Claire Cardie, Serge Belongie, Bharath Hariharan, and Ser-Nam Lim. "Visual prompt tuning." In European Conference on Computer Vision, pp. 709-727. Cham: Springer Nature Switzerland, 2022.

---

> > > ### Comment · Reviewer_4VnN · 2024-12-01
> > > **the advantage from cross-modality prompt transfer is clear**
> > >
> > > Given the authors' justification of the advantages of cross-modality prompt transfer, I adjust my score to a weak accept.

---

> > > > ### Author Response · Authors · 2024-12-02
> > > > **Thank you for your comments**
> > > >
> > > > Glad to know!
> > > >
> > > > We hereby express our sincerest gratitude for your valuable time and suggestions. They are of great help to our research.
> > > >
> > > > Wish you all the best!
> > > >
> > > > Authors of submission 2711

---

> ### Author Response · Authors · 2024-11-21
> **Response to questions 1 and 2**
>
> # **Question 1 (The training cost of our transferability metric)**
> We provide the training cost (measured in time, using an NVIDIA L40 48GB GPU with a CUDA version of 12.4) of our transferability metric here.
> Given a specific target task, we recorded the runtime (in seconds) for:
> + Training the universal projector for 100 epochs with different numbers of source tasks ($\mathcal{G}_T$ time).
> + Computing the MMD between a source prompt and a target task’s image embeddings ($\mathcal{G}_M$ time).
>
> Below are the results for $\mathcal{G}_T$ time:
> | Number of Source Tasks | 1 | 2 | 3 | 4 | 5 | 6 | 7 | 8 | 9 | 10 | 11 | 12 | 13 |
> | - | - | - | - | - | - | - | - | - | - | - | - | - | - |
> | L40 GPU Time (seconds) | 1.12 | 1.14 | 1.06 | 1.11 | 1.06 | 1.25 | 1.17 | 1.08 | 1.06 | 1 | 0.99 | 1.12 | 1.01 |
>
> Below are the results for $\mathcal{G}_M$ time:
> | Number of Source Tasks | 1 |
> | - | - |
> | L40 GPU Time (seconds) | 8.1 |
>
> In summary, given one target task, calculating the transferability between it and all the source tasks (13 in total) requires around $(8*13)+1=105s$ GPU time.
>
> We have 19 target tasks in total, so when we scale up the number of target tasks to 19, it requires $105*19=1995s$ GPU time, around half an hour using a NVIDIA L40 GPU to calculate the transferability between all 19 target tasks and 13 source tasks. This amount of time is negligible compared to performing prompt transfer on every source-target pair to brutally search for the best source prompts.
>
> The interesting part here is that calculating MMD takes much more time than training the universal projector. But we are sure that there are plenty of ways to reduce the compute load for calculating MMD. For example, our MMD was obtained by averaging 5 runs to ensure solid results. But in fact, 1 run is already enough. The time taken can be reduced 5 times if we calculate the MMD in only 1 run. If that, we only need $(2*13+1)*19=513s$ GPU time to calculate all the transferability values.
>
>
>
> # **Question 2 (the distribution of modality and task gaps in practice and a threshold to select the source prompt for transfer)**
>
> We put the ground-truth values of the gaps in our **Appendix D.2.5, page 10, Table 13**.
>
> Regarding a threshold to select the source prompt for transfer, in the current stage, we could not find it.
>
> We will exemplify this conclusion through a simple counter example:
>
> Below are the ground-truth gap values of target tasks Caltech101 and dSprites_ori:
> | | IMDB | SST2 | Laptop | Restaurant | Movie | Tweet | MNLI | QNLI | SNLI | Deontology | Justice | QQP | MRPC |
> | - | - | - | - | - | - | - | - | - | - | - | - | - | - |
> | Caltech101 | 1.116 | 0.759 | 1.284 | 1.269 | 1.320 | 0.900 | -0.141 | 0.239 | -0.180 | 1.053 | 1.110 | 0.082 | 1.121 |
> | dSprites_ori | 3.439 | 2.725 | 3.895 | 3.838 | 3.922 | 2.997 | 1.107 | 1.998 | 1.152 | 3.144 | 3.420 | 1.707 | 3.568 |
>
> On Caltech101, the source prompt MRPC is experiencing negative transfer (MRPC has a projection transfer performance of 88.26, whereas vanilla VPT has 88.73 on Caltech101) and has a gap value of 1.121.
>
> On dSprites_ori, SNLI has a similar gap value of 1.152. But SNLI experiences great positive transfer with a projection transfer performance of 42.35 compared to only 30.94 of VPT on dSprites_ori.
>
> This counter example shows that the actual values can only reflect the relative gaps between all the source tasks and a target task. Being greater or not, the values are only comparable within a target task. They cannot be extended to other target tasks.

---

> ### Author Response · Authors · 2024-11-21
> **Response to question 3 and 4**
>
> # **Question 3 (more concrete evidence of improvements in data-scarce tasks using cross-modality prompt transfer)**
>
> We here demonstrate the improvements through the comparison among Finetuning, Prompt Tuning, and our proposed Cross-Modality Prompt Transfer (CMPT):
>
> | | VTAB-Natural | VTAB-Specialized | VTAB-Structured |
> | - | - | - | - |
> | Finetuning | 75.88 | 83.36 | 47.64 |
> | Prompt Tuning | 77.09 | 80.85 | 45.45 |
> | Cross-Modality Prompt Transfer | **80.55** | **84.12** | **53.47** |
>
> This comparison shows:
> + The baseline VPT still underperforms finetuning on data-scarce tasks (VTAB-1K in our case). Given the fact that finetuning tends to overfit on data-scarce tasks [1], this demonstrates there is still a performance gap between VPT and finetuning.
> + Our proposed CMPT significantly boosts the performance of VPT, surpassing finetuning on every type of tasks.
>
> Note that we deem VTAB-1K to be a dataset containing 19 data-scarce tasks as every task only has 1000 training samples.
> The Finetuning performance is obtained from the original VPT paper [2].
>
> [1] Steiner, Andreas, Alexander Kolesnikov, Xiaohua Zhai, Ross Wightman, Jakob Uszkoreit, and Lucas Beyer. "How to train your vit? data, augmentation, and regularization in vision transformers." arXiv preprint arXiv:2106.10270 (2021).
>
> [2] Jia, Menglin, Luming Tang, Bor-Chun Chen, Claire Cardie, Serge Belongie, Bharath Hariharan, and Ser-Nam Lim. "Visual prompt tuning." In European Conference on Computer Vision, pp. 709-727. Cham: Springer Nature Switzerland, 2022.
>
>
> # **Question 4 (make the transferability metric more efficient)**
>
> In the current stage, based on our experiments and explorations, we did not find another way that can calculate the transferability faster and better compared to our current method.
>
> The only information we have about the source task is all condensed in the source prompts. And the same type of information about the target task shall be stored in the target prompts.
>
> Therefore, given the current literature of estimating task transferability based on prompts, quantifying the prompt similarity is a key point hard for us to bypass.
>
> However, in cross-modality scenarios, the gap between the source and target tasks is huge, making the source prompt and the target prompt lie in different semantic spaces. This just makes direct similarity calculation between prompts extremely unstable (as evidenced by Avg Cos and ON in Table 2 of our paper).
>
> Therefore, the source and target prompts need to be put in the same semantic space before any similarity calculation. Our proposed universal projector is designed for such a purpose. Plus, it does not introduce too much computation load. As discussed in question 1, to train the universal projector for a target task, even if we have 13 source prompts, the total amount of time is around only 1s GPU time, which is nearly negligible.
>
> In our future work, we will look deeper into the relations between source and target tasks, and hopefully we can come up with efficient and accurate prompt transferability estimation methods.
>  We sincerely hope that our response solves your questions! Thank you!

---

> ### Comment · Area_Chair_4uiP · 2024-11-25
>
> Dear Reviewer 4VnN,
>
> Could you kindly review the rebuttal thoroughly and let us know whether the authors have adequately addressed the issues raised or if you have any further questions.
>
> Best,
>
> AC of Submission2711

---

### Author Response · Authors · 2024-11-21
**Updates on the new revision**

Great thanks to all the reviewers for your valuable suggestions, based on which we have made the following modifications to our newly revised paper:

Main paper:

Added:
+ 1 Introduction: added a discussion on cross-modality prompt transfer hypothesis prior to introducing our work (line 53-54).
+ 2 Related Work: Clarify sentence (line 123-126).
+ 3.3 Attention Transfer: added a discussion on a similar approach in the literature (line 266-269).
+ Figure 3 and Table 1: explained how the relative performance is calculated.
+ 4 Experiments: added a discussion on the statistical significance of the improvements (line 393-394).

Deleted (The deletions are due to page limitation):
+ 3.3 Attention Transfer: The paragraph describing how attention operation works is deleted, as this shall be a convention.
+ 5 Conclusions: Moved future work from the main paper to the Appendix.

Appendix:

Added:
+ D.2.5: added the ground-truth gap values.
+ E: added a discussion section.

---

### Meta-Review · Area_Chair_4uiP · 2024-12-18

**Metareview:**

(a) This paper investigates cross-modality prompt tuning, demonstrating transferable text-based prompts, proposing a modality-aware transferability metric, and introducing attention transfer techniques that rival single-modality methods.

(b) Strengths: The paper presents a strong motivation and clear logic, with thorough experiments and detailed exploration of cross-modality prompt tuning. It introduces a novel perspective on prompt transfer, particularly focusing on modality and task gaps, and provides valuable insights for future research. The proposed attention transfer method effectively addresses modality challenges and enhances prompt reuse across domains. Extensive experimental evidence across multiple datasets shows that the approach can outperform recent benchmarks, demonstrating its potential, especially for data-scarce tasks. Overall, the paper is well-organized, easy to follow, and presents promising results.

(c) Weaknesses: In the initial version, the paper's claim that cross-modality prompt transfer boosts performance in data-scarce tasks is not convincingly supported by experiments. The advantages of cross-modality over single-modality methods remain unclear. The approach seems sensitive to hyperparameters, which could affect reproducibility, and there are cases where it underperforms, with its limitations not thoroughly discussed. The proposed transferability metric adds complexity and training costs.  The authors have addressed most weaknesses during the rebuttal.

(d) The most important reasons for acceptance are that the work introduces a novel perspective on prompt transfer, particularly focusing on modality and task gaps, and provides valuable insights for future research and the community.

**Additional Comments On Reviewer Discussion:**

(a) Reviewer 4VnN argues that the paper's claim that cross-modality prompt transfer improves performance in data-scarce tasks is not convincingly supported, with the advantage over single-modality methods and the complexity of the transferability metric remaining unclear. The authors have addressed the concerns, especially for the justification of the advantages of cross-modality prompt transfer. The reviewer increase the score to a weak accept.

(b) Reviewer mrx7 suggests that the paper would benefit from more comparisons with same-modality baselines, addressing the sensitivity to hyperparameters, and further discussion of the approach's limitations, especially when it underperforms. The authors have addressed most of the concerns and the reviewer decides to keep the score.

(c) Reviewer v8xH finds that the paper lacks clarity on the motivation for quantifying prompt transferability, the choice of the IN-21k pre-trained checkpoint, and the reasons for prompt transferability, with a deeper discussion on these points needed for better understanding. The authors have carefully addressed the issues and the reviewer tends to vote for acceptance.

(d) Reviewer Ve1v expresses concerns about the effectiveness of the proposed method, suggesting that the impact of extra parameters and source domain knowledge in the projection and attention transfer experiments is unclear, and recommends additional experiments with random source prompts, varied source data, different language models, visual prompt tuning methods, and more datasets to better demonstrate the method's effectiveness. The authors address all the concerns and the reviewer increases the score.

---

### Decision · Program_Chairs · 2025-01-22

Accept (Poster)